# Common seed dispersers contribute most to the persistence of a fleshy-fruited tree

Finn Rehling [1,2 ✉], Eelke Jongejans [3,4], Jan Schlautmann[1], Jörg Albrecht [5], Hubert Fassbender[1], Bogdan Jaroszewicz [6], Diethart Matthies [7], Lina Waldschmidt[1], Nina Farwig [1] & Dana G. Schabo[1]

Mutualistic interactions are by definition beneficial for each contributing partner. However, it is insufficiently understood how mutualistic interactions influence partners throughout their lives. Here, we used animal species-explicit, microhabitat-structured integral projection models to quantify the effect of seed dispersal by 20 animal species on the full life cycle of the tree *Frangula alnus* in Białowieża Forest, Eastern Poland. Our analysis showed that animal seed dispersal increased population growth by 2.5%. The effectiveness of animals as seed dispersers was strongly related to the interaction frequency but not the quality of seed dispersal. Consequently, the projected population decline due to simulated species extinction was driven by the loss of common rather than rare mutualist species. Our results support the notion that frequently interacting mutualists contribute most to the persistence of the populations of their partners, underscoring the role of common species for ecosystem functioning and nature conservation.

[1] University of Marburg, Department of Biology, Conservation Ecology, Marburg, Germany. [2] University of Marburg, Department of Biology, Animal Ecology, Marburg, Germany. [3] Radboud University, RIBES, Nijmegen, Netherlands. [4] NIOO-KNAW, Department of Animal Ecology, Wageningen, Netherlands. [5] Senckenberg Biodiversity and Climate Research Centre Frankfurt, Frankfurt, Germany. [6] University of Warsaw, Faculty of Biology, Białowieża Geobotanical Station, Białowieża, Poland. [7] University of Marburg, Department of Biology, Plant Ecology, Marburg, Germany. ✉email: finn.rehling@nature.uni-freiburg.de

Mutualisms are beneficial for each partner, shape the coevolution of species, and contribute to the functioning of ecosystems[1]. A main goal of conservation is to maintain networks of mutualistic interactions, with the ultimate aim of conserving biodiversity and ecosystem functioning[2]. Interacting organisms, however, are expected to differ in their contribution to each other's fitness and to ecosystem functions[3]. In order to identify and protect key species in ecosystems, scientists aim at quantifying the functional outcome of mutualistic interactions[3,4]. Quantifying the long-term outcomes of mutualisms, however, is difficult due to their strong context-dependency, limiting our understanding of the functional role of species for the maintenance and conservation of biodiversity and ecosystems[5].

The seed dispersal mutualism is an important ecosystem process that contributes to animal nutrition and the plant's regeneration cycle[6–8]. In return for fruit pulp[9,10], animals deposit seeds in favorable microhabitats, improve seed germination, and help plants to colonize new locations[11–15]. At larger scales, seed dispersal maintains meta-community dynamics[16] and helps plants to migrate[17,18]. Seed dispersal by animals is effective and beneficial for the plants that are dispersed. However, to date there is only indirect evidence for the long-term benefits of animal seed dispersal for plants. Such benefits have been deduced, for example, from the differences between the spatial genetic structure of parental plants and that of their offspring[19,20], from the disruption of plant regeneration after animal dispersers had become extinct[20–23], or from the results of trait-based modeling of seed dispersal[24,25]. Direct investigations of the effects of seed dispersal

by animals across all stages of the life cycle of plants are rare and so far the functional role of a maximum of five disperser species has been investigated[26–32]. This is mainly due to the difficulty of linking the behavior of animals to their cascading effects on the populations and demography of plants, whose lifespan can be decades to several millennia[33].

A potential solution for understanding the overall effect of seed dispersal on plants is offered by studying the seed dispersal loop[34,35] (Fig. 1). By following the fate of animal-dispersed seeds through space and time it is possible to break down the complex seed dispersal mutualism into individual processes of the life cycle of a plant that can influence the outcome of the mutualism (Fig. 1a). The loop starts with the visitation of fruiting plants by animals and the removal of fruits, followed by the transportation of seeds and their deposition, followed by seed germination, establishment of the seedlings and by their development into adults. These consecutive steps can be linked using stage-structured population models to quantify the overall effect of seed-dispersing animals on plant populations.

The overall effect of a certain seed-dispersing animal species on a plant population has been called seed dispersal effectiveness (SDE)[3]. The probably most comprehensive measure of SDE is the effect of seed dispersal on the population growth rate of a plant. The SDE can be quantified as the product of the quantity and the quality of seed dispersal. The quantity of seed dispersal is equal to the frequency of interactions between animals and plants. The interaction frequency of seed dispersers can be expressed as the number of visits to a plant by an animal, multiplied by the number

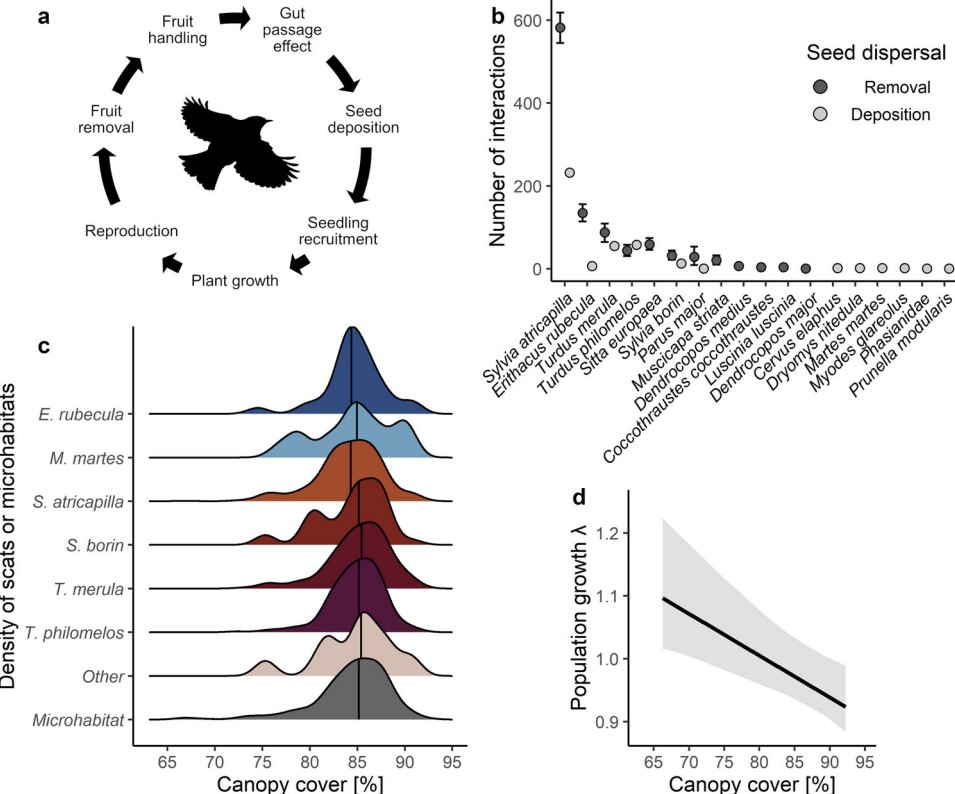

**Fig. 1 The seed dispersal loop[34] of animal-dispersed plants. a** Animal seed dispersal as a demographic bridge between reproductive adults and new plant recruits. **b** Fruit removal: the number of visits of animal species to the fleshy-fruited tree *Frangula alnus* ('Removal'[51]) and the number of scats with seeds of *F. alnus* ('Deposition'[53]) in the Białowieża Forest, Eastern Poland. Two more disperser species were only observed using camera traps (Supplementary Table 3). Means ± 95% prediction interval based on 500 bootstraps of dispersal data. **c** Seed deposition: the effect of different animal species on relative scat deposition density along the canopy cover gradient. The relative availability of microhabitats is shown for comparison. A vertical line indicates the median of a distribution. **d** Plant performance: relationship between the population growth rate of *F. alnus* and canopy cover when seeds are solely dispersed by gravity. The 95% prediction interval is based on 500 bootstraps of plant data. The bird silhouette (public domain) was obtained from phylopic.org.

of seeds dispersed per visit. The quality of seed dispersal describes the effect of dispersal on the fate of a seed until it develops into an adult plant. Analyzing the effects of different animal species on the individual processes of the seed dispersal loop may help to identify the demographic processes and mutualistic partners that contribute most to the persistence of plants[3,34,35]. However, there are no studies on the seed dispersal loop in species-rich and complex ecosystems[36]. Thus, it remains unclear whether the functional role of animals is context-dependent[3,27,37,38], can be predicted based on the interaction frequency of animals with plants[4] or based on ecological traits[39,40], and whether the loss of interactions due to the extinction of individual species can be compensated by other animals[12,15,41]. Even when all mutualistic dispersers are lost simultaneously, plants may still persist, if seed dispersal by gravity, wind or water is sufficient[42,43]. Understanding the long-term consequences of seed dispersal is required to gain insights into the coevolution of plants and animal mutualists[44–46], and should be used to improve conservation management decisions[47,48].

In this study, we explored long-term effects of the seed dispersal mutualism between 20 frugivore species (14 birds and 6 mammals) and the population of the mid-successional tree *Frangula alnus* (glossy buckthorn)[49] in the primeval Białowieża Forest[50], Poland (Supplementary Tables 1–3, Supplementary Fig. 1). We used integral projection models (IPMs) to study the effect of animal seed dispersal on the full life cycle of *F. alnus* along the gradient of canopy cover in the forest (Fig. 1, Supplementary Fig. 1). The IPMs were partly built on datasets of previous studies: in a first step, we observed with binoculars the removal of fruits, and the fruit handling behavior (i.e., removing or dropping fruits, and crushing seeds) by animal species over 936 h on 52 reproductive individuals of *F. alnus* (Fig. 1b)[51,52]. In a second step, we determined animal-specific deposition patterns of seeds by collecting and DNA barcoding 1729 scat samples with 9590 seeds of *F. alnus* and other fleshy-fruited plants along the forest canopy gradient (Fig. 1c)[15,53]. In a third step, we included an estimate of seed germination after gut passage from a recent meta-analysis in the models[11]. In a fourth step, we investigated seedling recruitment by experimentally sowing 2500 seeds in the forest[15]. These comprehensive datasets were extended in this study by investigating the growth, survival, and reproduction of 938 individuals of *F. alnus* along the forest canopy gradient over three years (Fig. 1d). Our 10 years of research on seed dispersal in one ecosystem allowed us to seek answers to the following questions: (i) Do populations of animal-dispersed plants need animal dispersal to persist? (ii) What are the key factors determining the effectiveness of animal dispersers? (iii) Can interactions of extinct species be functionally compensated by the remaining animal community?

By analyzing the whole life cycle of a fleshy-fruited tree, this study shows that plant population growth is increased by the seed dispersal by frugivorous animals compared to seed dispersal by gravity, especially when trees need to colonize forest gaps. The interaction frequency of seed-dispersing animals was found to be a suitable predictor for the total effect of the animals on plant demography. Our results support the notion that seed-dispersing animal species in temperate forests may be largely redundant in the quality of the services they provide. Seed dispersers could potentially take over the ecological role of other species if they quantitatively compensate lost interactions. This study emphasizes the benefits of animal seed dispersal for plants, and the role of common seed dispersers for ecosystem functioning.

## Results and discussion
### Fruit removal and seed deposition.
We recorded 20 animal species (14 birds and 6 mammals) acting as seed dispersers of

*F. alnus* in the Białowieża Forest[51–53]. We estimated the interaction frequency of animal species with *F. alnus* by multiplying the number of their visits × probability of handling a fruit during a visit × mean number of handled fruits. However, eight of the animal species were not observed removing seeds, and identified as dispersers only by DNA-barcoding scats or by camera trapping (Supplementary Table 3). To harmonize the complementary seed disperser information obtained by the different methods[54], we assumed that the effects of these eight animal species were functionally similar to those of other rare dispersers of *F. alnus* for which data from seed removal observations were available (see "Methods"). We found that the relative interaction frequency of animals with *F. alnus* was highly variable. Four bird species accounted for 86.6% (95% confidence interval: 82.4–89.8%) of the interactions: *Sylvia atricapilla* (58.7%, 53.2–63.9%), *Turdus merula* (15.0%, 9.8–20.6%), *Erithacus rubecula* (8.3%, 6.2–10.4%), and *T. philomelos* (4.6%, 2.9–6.5%), while each of the sixteen remaining animal species contributed ~1% or less to total fruit removal (Supplementary Table 3, Supplementary Fig. 2).

Only a subset (10.3 %) of the collected scats contained seeds of *F. alnus*. Most of these scats were from the birds *S. atricapilla* (n = 232), *T. merula* (n = 55), and *T. philomelos* (n = 58), while another 30 scats were from the remaining animal community (Fig. 1c, Supplementary Table 3). The relatively low sample size of scats with seeds of *F. alnus* allowed us to predict seed deposition of *F. alnus* for only three of the 20 studied disperser species. Therefore, we used information from all scats containing seeds of fleshy-fruited shrubs and trees (n = 1729; see "Methods") to predict the deposition of scats with seeds of *F. alnus* by individual disperser species. More than 30 scat samples were available for six animal species; all species with fewer scat samples were pooled as 'other'. In the transects where we collected scats, we also took hemispherical photos with a fish-eye lens to determine the proportion of area covered by canopy[55]. This allowed us to link the deposition of seeds by the dispersers to the microhabitat in which seeds potentially produce seedlings after dispersal (Fig. 1c).

We found that *T. merula*, *T. philomelos*, and *E. rubecula* dispersed only 3–4% of the seeds to the 50%-brightest environments along the canopy gradient. We assumed that the frequency distribution of microhabitats along the study transects was representative for the entire forest. The three birds deposited seeds less often in bright environments than expected by chance (i.e., 7.3% of available area). In contrast, seed deposition along the canopy cover gradient by other animals was indistinguishable from random dispersal (Fig. 1c, Supplementary Figs. 3, 4). Our results revealed that seed dispersal by several important disperser species in the Białowieża Forest is non-random not only for the plant community as a whole[15], but also for the seeds of individual species. Non-random seed dispersal by animals seems to be common[12,14] and has been observed, for instance, for bellbirds[56], muntjacs[26], and lemurs[57] along canopy gradients in tropical forests.

### Vital rates and population growth of *F. alnus* along the gradient of canopy cover in the forest.
To assess how seeds deposited along the canopy cover gradient contribute to population growth of *F. alnus*, we collected data on seedling recruitment by sowing 2500 seeds into 40 plots at four sites and following their fate over three years (see "Methods"). In addition, we recorded the survival, growth, reproduction, breakage, and re-sprouting of 938 individuals of *F. alnus* at 14 study sites (see "Methods"). Based on the analyses of GLMMs (R-package *glmmTMB*[58]) in R[59], we found no effects of increasing canopy cover on seedling recruitment (Supplementary Discussion 1, Supplementary Fig. 5), but strong negative effects on the survival, growth, and reproduction

of *F. alnus* (Supplementary Tables 4, 5, Supplementary Figs. 6–9). Moreover, survival, growth, and reproduction of *F. alnus* were positively related to plant size, but some of these relationships differed among study years (Supplementary Tables 4, 5, Supplementary Figs. 6–9). However, the effect of the interaction between plant size and canopy cover on vital rates was not part of the most parsimonious model (Supplementary Table 4). We then integrated these regression models into microhabitat-structured integral projection models (IPMs). IPMs are mechanistic models, which allow to scale up observations on single individuals to the population level. By modeling ecological factors influencing vital rates (i.e., survival, growth, and reproduction), IPMs can be used to predict the dynamics and growth of populations influenced by factors of interest[60,61]. Here, the state of the population of *F. alnus* in the IPMs was described simultaneously by the size distribution of plants (Supplementary Fig. 10) and the location of plants along the canopy cover gradient (see "Methods").

We first quantified the effect of local canopy cover on population growth of *F. alnus* (Fig. 1d). For this, we modeled the population growth of *F. alnus* at several points along the canopy gradient and assumed that there was only gravity dispersal and all seeds would thus be deposited in the same environment as their parents. Population growth gradually declined with increasing canopy cover (Fig. 1d), from being positive ($\lambda = 1.16$, 95%-prediction interval: 1.02–1.22) in bright environments (canopy cover = 66.3%), to neutral ($\lambda = 1.00$, 0.96–1.08) at sites with intermediate conditions (canopy cover = 79.9%), to negative ($\lambda = 0.95$, 0.89–1.00) in the closed forest (canopy cover = 90.7%). These results characterize *F. alnus* as a gap-dependent, successional tree of temperate forests, in line with previous studies[32,62]. We then used the microhabitat-structured IPM with gravity dispersal (Fig. 1d) to weight the population growth rates according to the relative abundance of the microhabitats along the canopy gradient in the forest (Fig. 1c). The microhabitat-structured and -weighted IPM predicted that a population of *F. alnus* whose seeds are only dispersed by gravity would decline by 3% each year ($\lambda = 0.97$, 0.94–1.04).

**Effects of animal seed dispersal over the full life cycle of *F. alnus*.** In a next step, we combined data on plant vital rates, fruit removal and seed deposition in animal species-explicit microhabitat-structured IPMs to model the effect of seed dispersal by animals on the *F. alnus* population. We increased the proportion of seeds dispersed by animals in the IPM from 0% to 100% (see "Methods"). We analyzed two scenarios. In the first scenario, we assumed that *F. alnus* occurred along the whole gradient of canopy cover ('fully established' in Fig. 2), as observed in the studied forest. The higher the proportion of seeds dispersed by animals was, the higher was the population growth rate. When all seeds were dispersed by animals, the population would be constant in size ($\lambda = 1.00$), i.e., an increase in $\lambda$ by $0.025 \pm 0.001$ in comparison to a situation in which all seeds were dispersed by gravity. In the second scenario, we assumed in the IPM that *F. alnus* only occurred in the closed forest and relied on seed dispersal by animals to reach the 50%-brightest microhabitats ('gap colonization', Fig. 2). As in the first scenario, animal dispersal increased population growth of *F. alnus* up to $\lambda = 1.00$ if all seeds were dispersed by animals. However, the positive effect of animal dispersal in comparison to seed dispersal by gravity ($\Delta\lambda = 0.029 \pm 0.001$) was 13.7% stronger than in the first scenario. The beneficial effect of seed dispersal by animals in both scenarios was strongly related to the positive effect of the gut passage on seed germination (+70%)[11] in the IPMs.

Our results show that animal seed dispersal can be especially beneficial for populations of *F. alnus* if new habitats are colonized.

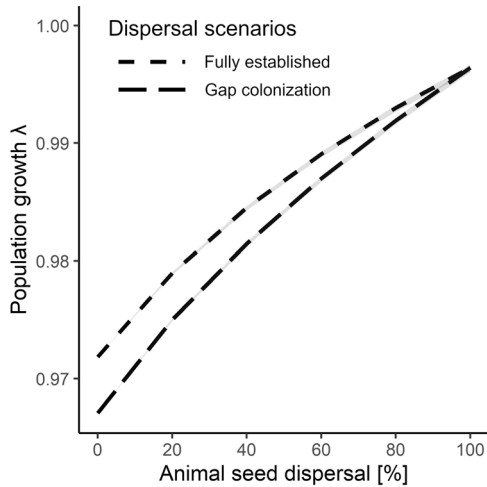

**Fig. 2 Benefits of animal seed dispersal.** The effect of seed dispersal by 20 animal species on the population growth rate of *F. alnus* in Białowieża Forest, Eastern Poland. Two scenarios are presented: (1) Plant individuals occur along the entire canopy gradient ('fully established', short-dashed line) or (2) occur only in closed forest (the 50%-darkest microhabitats, i.e., 92.7% of available area) and depend therefore on animal seed dispersal for establishment in forest gaps ('gap colonization', long-dashed line). In a 'fully established' population occurring along the full canopy gradient, gravity dispersal and fruits dropped by animals can result in successful plant regeneration where the canopy is not closed. In comparison, the population growth rate is reduced in a 'gap colonization' population if gravity dispersal is predominant, because plant establishment and growth is much reduced under a closed canopy. When 100% of seeds are dispersed by animals, the long-term stable stage distribution of plants and their distribution along the canopy cover is identical, and so is the population growth rate $\lambda$. Prediction intervals (95%) based on 500 non-parametric bootstraps of dispersal data were calculated for the effect of the proportion of seeds dispersed on $\lambda$. Please note that uncertainty in the effect of seed dispersal resulted in only extremely small differences in the population growth rate of *F. alnus*.

However, our findings may even underestimate the value of animal seed dispersal for the long-term persistence of plants in dynamic forests[63,64]. We studied the population dynamics of *F. alnus* only over the course of three years and the canopy structure was held constant in the IPM. When forest succession leads to the competitive exclusion of *F. alnus* by late-successional trees, seed dispersal by animals to newly created forest gaps is essential for the long-term persistence of *F. alnus*. The results confirm the conclusions based on natural history observations about the importance of the seed dispersal mutualism between plants and animals for population persistence and forest dynamics[6].

**Interaction frequency as an estimate of seed dispersal effectiveness.** To analyze how individual animal species influence the population growth of *F. alnus*, we calculated the seed dispersal effectiveness (SDE) of different animals[3]. As a measure for SDE, we used the change in population growth following the loss of interactions with an animal disperser (i.e., interaction deficit after species loss). We found that the loss of seed dispersal by some animal species reduced the population growth rate of *F. alnus* (Fig. 3a): The reduction was strongest when either *S. atricapilla* (−47.2%), *T. merula* (−10.1%), *E. rubecula* (−5.5%), or *T. philomelos* (−3.0%) were lost. In contrast, the extinction of each of the 14 other animal species reduced the population growth of *F. alnus* by less than 1% (representing together 33% of the entire studied frugivore community and 70% of the dispersers of *F. alnus*).

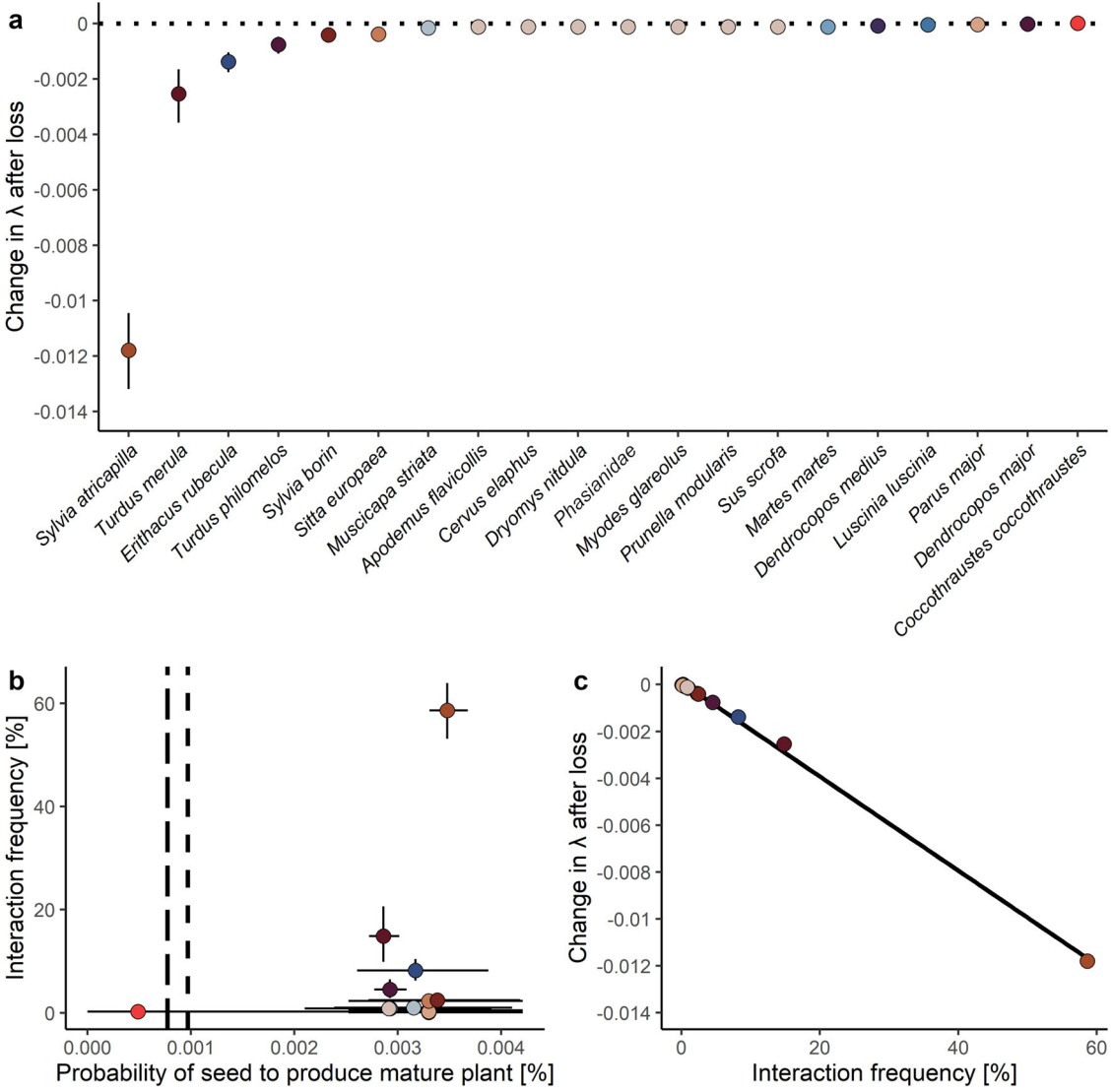

**Fig. 3 Total effectiveness of seed dispersal by animals. a** Projected consequences of the extinction of each of 20 frugivorous animals on the population growth rate of *F. alnus* assuming complete interaction deficit (i.e., the interactions with an animal are lost completely). Animal species are in order of declining importance as dispersers for *F. alnus*. Animal species depicted by the same color ('other' frugivores, see Supplementary Table 3) were assigned the same values for seed removal and for seed deposition, resulting in the same estimate for seed dispersal effectiveness. **b** The 'landscape' of seed dispersal effectiveness of animal species and gravity dispersal for *F. alnus*. The y-axis represents the quantity of seed dispersal (i.e., the relative interaction frequency of each animal species with *F. alnus*) and the x-axis represents the quality of seed dispersal (i.e., the probability of a seed to produce a mature plant when dispersed by a certain animal species). The two vertical lines depict the consequences of exclusive gravity dispersal in the two scenarios shown in Fig. 2 for seed dispersal quality. **c** The relationship between seed dispersal effectiveness (i.e., change in population growth after interaction loss) and the relative interaction frequency of different animal species with *F. alnus*. Color coding in (**b**) and (**c**) is the same as in (**a**) and Fig. 1c. Mean ± 95% prediction intervals based on 500 bootstraps of dispersal data.

We then divided the SDE of animal species into the contributions of the quantity and the quality of seed dispersal[3] (Fig. 3b). To study the quantity component, we used the relative interaction frequency of each animal species with *F. alnus*. To study the quality component, we calculated the probability of a seed to produce a mature plant after being dispersed by a particular animal species (Supplementary Fig. 11). The quality of seed dispersal is the result of fruit handling and gut passage of the seed on seed germination[11], seed deposition by animals along the canopy gradient, and plant growth to adulthood (Fig. 1). We found that seed dispersal quality significantly differed among animal species (Fig. 3b). SDE of an animal species was strongly related to the quantity (Fig. 3c, Spearman's $\rho = 0.99$, $p < 0.001$) but much less strongly to the quality of seed dispersal (Spearman's $\rho = 0.21$, $p = 0.662$).

The strong relationship between SDE and the interaction frequency of animals with *F. alnus* may have been due to the following not mutually exclusive reasons: (1) small differences among species in patterns of seed deposition, (2) small effects of canopy cover on seedling recruitment, (3) small effects of density dependence on seedling recruitment, (4) low predation of seeds, (5) the assumption that passage through the gut of all dispersers had the same effect on seed germination.

(1) We found only small differences in seed deposition among animals. This may be due to the fact that the populations of the various disperser species have similar movement patterns in Białowieża Forest[15]. In contrast, strong differences in habitat use among animals may lead to

striking differences in seed deposition[12,14,26,56]. To test the sensitivity of seed dispersal quality to variation in seed deposition, we simulated unrealistic extreme patterns of seed deposition. For example, we simulated that animals dispersed seeds exclusively to bright environments (Supplementary Fig. 12). These extreme deposition patterns increased the differences in seed dispersal quality between main dispersers from two-fold to 23-fold. However, differences in the quantity of seed dispersal still were much larger (972-fold) than those in seed dispersal quality.

(2) The effect of differences in canopy cover on seedling recruitment and survival was rather small in comparison to what has been found in other type of habitats. First-year survival of seedlings of *F. alnus* decreased from c. 60% in bright environments (canopy cover = 66.3%) to 20% in closed forest (canopy cover = 90.7%). In contrast, in environments that result in very heterogeneous seed or seedling survival, the effectiveness of seed dispersal can depend strongly on the few animal species that disperse seeds to favorable microhabitats[27]. For example, in arid environments such as deserts, most seedlings survive only in the shade of nurse plants where they are protected from heat stress[65], while seeds dispersed by gravity or deposited by animals in open areas do not contribute to plant regeneration[27] (but see refs. [66,67]).

(3) It has been frequently suggested that seed dispersal by animals is especially important for escaping the increased mortality close to conspecific plants[22,68,69]. As animals differ in their behavior and physiology, some animal species deposit seeds more often beneath conspecific plants than others, resulting in pronounced differences in seed dispersal quality among animals[70,71]. In line with this, we found that inefficient seed dispersal by animals beneath conspecific adults of *F. alnus* along the canopy gradient was frequent (c. 63%), but differed among species (Supplementary Discussion 2, Supplementary Fig. 13). However, we did not account for density-dependence in the IPM, because no density-dependence has been found in *F. alnus*[72] (Supplementary Discussion 2). Furthermore, secondary seed dispersal by ants or water in riparian forests, such as the ash-alder forest in our study area, is likely to reduce the occurrence of density-dependence[42].

(4) Seed predation by animals may strongly limit the quality of dispersal[38,73]. However, most animals rarely (or not at all) crush seeds of *F. alnus*. The low predation by animals contributed to the small differences in the quality of seed dispersal among animal species. The only exception was *Coccothraustes coccothraustes* which predated 80% of seeds handled and negatively affected population growth of *F. alnus* (Fig. 3).

(5) A recent, extensive meta-analysis of over 2500 experiments showed that the positive effect of a passage through the gut of animals on seed germination is very strong for species in temperate regions (+70%)[11]. This effect was mostly due to the improved germination after the removal of pulp during fruit consumption. As gut passage effects on seed germination do not strongly differ among animal species[11], we used the same parameter value for different animal species in the IPMs. Thus, ultimately, the quality component with the largest effect on seed dispersal effectiveness in our study did not differ among animal species. Including variation in gut passage effects on seed germination in the IPMs would result in stronger differences in the quality of seed dispersal among species, but not to an extent that outweighed differences in interaction frequency[74].

Together, these factors may explain why differences in interaction frequency between disperser species are often more pronounced than differences in seed dispersal quality[4]. Even if a disperser is not very effective on a per-interaction basis, e.g., because it disperses only a small proportion of seeds into suitable microhabitats, this can be numerically overcompensated if it frequently interacts with the plant[4]. Thus, the common bird species (e.g., *S. atricapilla* and *T. merula*), which are among the most abundant bird species in European temperate forests[75] and most frequently interact with *F. alnus*, have the strongest impact on its population growth. These bird species are also quantitatively the most important seed dispersers for other fleshy-fruited plant species in Białowieża Forest[51] and throughout Europe[17]. In contrast, many of the animal species with a negligible effect on plant population growth were forest specialists or rare. As these species are likely to be lost first in case of negative anthropogenic impacts, seed dispersal may be relatively robust to species loss[52].

**Potential for interaction compensation among animals.** Animal species that are currently not functionally important for seed dispersal, may become important under future conditions if they take over the role of declining disperser species. In a second scenario, we tested if the animal community can keep up the effectiveness of seed dispersal when one of the four common disperser species goes extinct, and the remaining community compensates the reduced fruit removal (i.e., interaction compensation of extinct species). Our simulations showed that animals would be fully able to quantitatively compensate the interactions of lost dispersers in the IPM, because morphological mismatches between plants and their dispersers rarely occur at the species-level in small-fruited plants[76,77].

In this scenario, we found that the potential extinction of the main dispersers of *F. alnus* (*S. atricapilla*, *T. merula*, *E. rubecula*, and *T. philomelos*) was well buffered by the remaining animal community (Fig. 4, <4% functional loss). This was due to the small differences in the quality of seed dispersal among mammalian and avian dispersers observed before. This indicates that there might be no disperser of *F. alnus* whose contribution to population growth is unique[78–80]. A high redundancy of seed

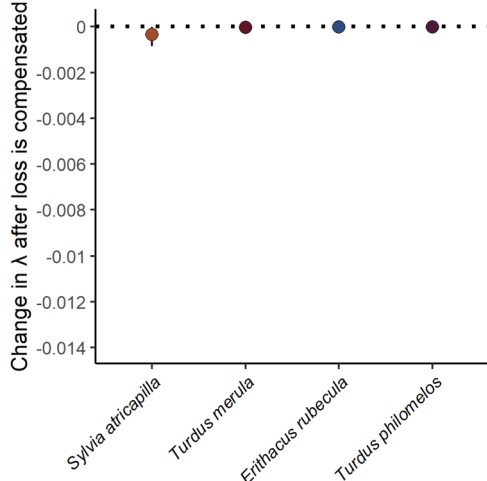

**Fig. 4 Interaction compensation after frugivore loss.** Projected consequences of the extinction of each of the four main frugivorous animals on population growth of *F. alnus* assuming complete interaction compensation after their loss (i.e., a species is lost, but the remaining animal community compensates interactions). Mean ± 95% prediction intervals based on 500 bootstraps of dispersal data.

dispersal might be a useful mechanism to buffer the large variation in the abundance of individual animal species among years or environments[81,82]. It further indicates that interaction rewiring by introduced species in degraded habitats may not only lead to structural[83], but also to functional compensations.

However, we did not investigate how changes in the abundance of a particular species influence the abundance, behavior, and seed deposition of other animals[84]. Direct and indirect effects of changes in the environment and the ecological interactions associated with species loss are likely to disrupt seed dispersal services[15,48]. Many of the disperser species with weak effects on population growth of *F. alnus* are specialized in a different type of diet than fruits (e.g., insects). Mostly the four main dispersers consume significant amounts of fruits, as they switch their diet to fruit resources once they are available[85,86]. In addition, before species are lost, the environment has changed in its structure and with it the effectiveness of ecological interactions[15,87]. Therefore, dietary preferences of animals and cryptic functional loss of interactions, rather than morphological mismatches[76], can be expected to limit the ability of the remaining species to compensate interactions after the loss of common species in our study system. Moreover, the loss of certain animals could affect population processes that are beyond the scope of this study, for instance range expansion[32,88], gene flow[89], and plant migration in response to climate change[17,18].

In recent decades, over 550 million birds have been lost in Europe[75,90]. While populations of habitat generalists and birds associated with agricultural landscapes declined the most, forest species also declined, but to a smaller extent[75,90,91]. Populations of the three key dispersers in this study, *E. rubecula* (+21.9 million), *S. atricapilla* (+54.9), and *T. merula* (+29.2), have increased over the same period[75]. The findings of this study on the interaction frequency of animal dispersers may thus already reflect the historic compensation of interactions that has taken place due to shifts in the animal community in recent decades. Yet, the three key dispersers are migratory, and are illegally hunted in the Mediterranean Basin[92]. A lack of international coordination and on-ground implementation of the conservation of migratory species[93,94] renders the seed dispersal mutualism of *F. alnus* in Białowieża Forest vulnerable to anthropogenic pressures.

**Conclusions**. Projected over the life cycle of *F. alnus*, seed dispersal by animals has positive effects on its population growth. The seed dispersal mutualism is relatively robust to the loss of single disperser species, because most animal species only consume few fruits which makes them functionally less relevant. However, if common seed dispersers decline, the effectiveness of seed dispersal will strongly decrease when the remaining community is not able to quantitatively compensate the ecological interactions[95]. We further conclude that differences in interaction frequency between mutualists may typically be more pronounced than differences in the effects of interaction quality on partners. Therefore, the interaction frequency may be a suitable surrogate for the total effect of mutualists on their partners[4,96]. This provides an empirical base for large-scale analyses of quantitative interactions in evolutionary, network, and trait-based ecology[97–99]. The particular importance of the frequency of an interaction for its overall effectiveness found in this study likely holds true also for other types of mutualistic interaction (e.g., pollination, pest control, ant-plant interactions)[100,101]. This highlights the role of frequently interacting and common species for ecosystem functioning across spatial and ecological scales[102]. To stop and reverse the ongoing decrease in the abundance of animals, especially that of common species, is thus key for

reinforcing the multifunctionality of ecosystems and conserving biodiversity.

## Methods

**Study area and sites**. Our study took place in the Białowieża Forest (Fig. 5) which covers an area of c. 1500 km² across the border of Poland and Belarus. At present, the 630 km² of forest in Poland are divided into the Białowieża National Park (c. 105 km²) and forests managed by state forestry. In an area of about c. 48 km² of the Białowieża National Park, human interference has been minimal for over half a millennium and that part has been strictly protected since 1921, making it the best-preserved lowland forest in Europe. In contrast, since the First World War, commercial logging has shaped more than 80% of the Polish part of the forest that is not the national park[50,103]. The ash-alder floodplain forests (*Fraxino-Alnetum* community) of the Białowieża Forest are home to a diverse community of at least fifteen woody, fleshy-fruited plant species and at least 41 frugivorous animal species. The frugivore community consists of small-bodied passerines (e.g., *Sylvia atricapilla*, *Erithacus rubecula*, *Turdus merula*), forest specialists (e.g., *Tetrastes bonasia*), and mammals of different size (e.g., *Dryomys nitedula*, *Martes martes*, *Bison bonasus*)[51,53,76].

All sampling took place at 17 study sites in ash-alder floodplain forests in both the managed (stand age: c. 70 years, $n = 11$) and the old-growth part (stand age: c. 100–150 years, $n = 5$) of the Białowieża Forest. Due to logistic constraints, we assessed each process important for the demography of *F. alnus* only at subsets of all study sites (Fig. 5, Supplementary Tables 1, 2): seed removal (at 15 sites), seed deposition (12 sites), seedling recruitment (4 sites, of which each site consisted of 10 plots, each with three subplots) and plant demography (14 sites). Our sites were distributed over an area of c. 400 km², i.e., two-thirds of the Polish part of the Białowieża Forest.

**Study species**. *Frangula alnus* Miller (Rhamnaceae) is distributed from Morocco throughout most of Europe to western Asia[104]. It grows as a shrub or as small tree in open environments or in the understorey of mid-successional forests[49]. In late-successional forests, shade-tolerant plants outcompete *F. alnus*[62], but its growth and regeneration may continue in canopy gaps[105]. In Białowieża Forest, *F. alnus* produces fruits from the end of July to October. The black fruits have a diameter of 6.5–10.7 mm and contain on average two seeds with a mass of 21.2 mg (range 10.3–36.0 mg)[76]. Seeds are primarily dispersed by small birds and mammals, and secondarily by ants or water[42]. They are physiologically dormant and both light

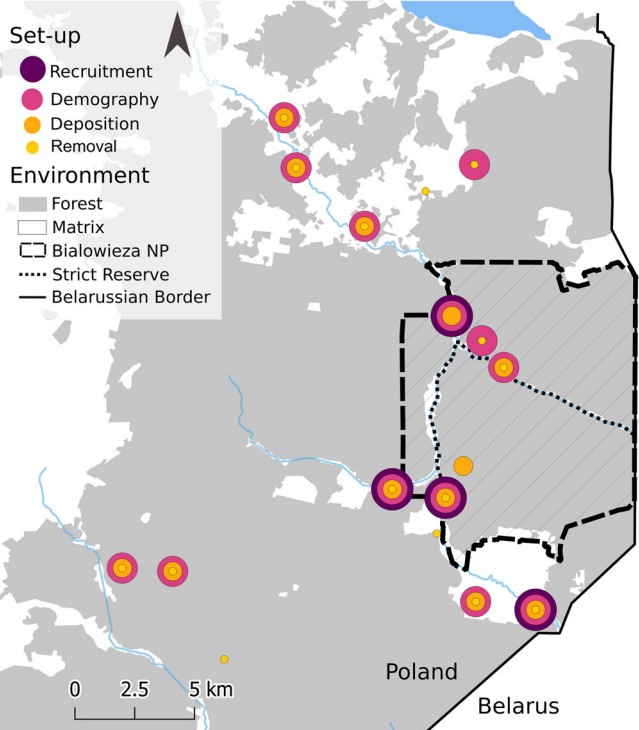

**Fig. 5 Study area and sites.** Map showing the location of the 17 study sites in the Białowieża Forest, Eastern Poland, and the studies conducted at these locations. The map was based on OpenStreetMap[113] and created using QGIS[114].

and cold-stratification improve germination[49]. *Frangula alnus* can produce clonal side-trunks and has the ability to re-sprout after breakage[49].

**Seed removal.** To quantify the interaction frequency of animal species with *F. alnus*, trained field staff (17 observers) recorded the seed removal and fruit handling by frugivores at 15 sites during the fruiting period in 2011 and 2012. Depending on the availability of fruiting individuals at the study sites, we selected one (at 2 sites), two (at 7 sites), or three (at 12 sites) reproducing individuals of *F. alnus* per year, 52 individuals in total (Supplementary Tables 1, 2). Frugivorous mammals and birds visiting these individuals were observed with binoculars from camouflaged tents on three separate days for a period of 6 h starting at sunrise, i.e., for in total 936 h of observation. For each frugivore species, the number of visits, the number of fruits eaten during each of these visits, and details of their fruit-handling were recorded. We differentiated between three types of fruit-handling: (i) swallowing or removing and (ii) dropping of fruits, and (iii) crushing of seeds. By differentiating between the swallowing and the crushing of seeds (see also 'integral projection models'), we accounted for the potential negative effects of animals that can act as seed predators or dispersers (e.g., *C. coccothraustes*). Rarely, we also observed the pecking of fruits ($n = 22$)[76]. However, the pecking of fruits usually served an exploratory probing of fruits and did not result in the removal of fruits. Thus, in this study, we categorized the pecking of fruits as a disperser visiting a reproductive tree, but not successfully handling a fruit. If groups of frugivores were visiting a tree at the same time, we recorded the number of visits and focused only on the behavior of one individual. Overall, 1006 frugivore visits were observed and whether frugivores handled fruits or not was successfully observed in 766 of 821 cases (93.3%). For further details on the methods for observing fruit removal, see the original study[51] that involved *F. alnus*.

**Seed deposition.** To quantify seed deposition patterns by the animal species along the canopy gradient in the forest, we collected scats of frugivores containing seeds of any species of the associated fleshy-fruited plant community at 12 study sites from 2016–2018. At each study site, we set up five 100 m transects which were at least 20 m apart. Along each transect, we searched for animal scats within 1 m wide strips to the left and right of each transect, covering a total area of 1000 m² per study site. The transects were checked every ten days during the fruiting season of the plant community lasting from mid-June to mid-October. After heavy rains, scat collection was paused for two days. At each study site, we collected scats eleven times in 2016 and 2018 and nine times in 2017 due to a shorter fruiting season. We collected all bird scats with seeds of the entire fleshy-fruited plant community to identify the frugivore in the laboratory using DNA barcoding. Mammal scats were assigned visually to species in the field. Seeds from mammal scats were counted in the field or also collected for genetic identification. The seeds from the scat samples were stored at −20 °C on the same day until they were used for frugivore identification in the laboratory.

To identify the frugivore species that had deposited the scat, we followed the DNA barcoding protocol of ref. [14]. DNA extraction and the PCR amplification took place in the Conservation Ecology laboratory of the University of Marburg (Germany). DNA samples were then sent to LGC Genomics (Berlin, Germany) or Macrogen Europe (Amsterdam, Netherlands) for DNA purification and sequencing. The final DNA sequences were edited with CodonCode Aligner (Version 9.0.1, CodonCode Corporation) and species were identified using the Barcode of Life identification system (BOLD)[106]. For the analysis we only used samples whose sequences had a >98% similarity with recorded sequences in BOLD. We successfully identified the frugivore species for c. 90% of our samples. Although we apparently found scats deposited by 'Meleagris gallopavo', we assessed this as unrealistic because it does not occur in Białowieża Forest and did not include the disperser at species level ('Phasianidae'). For further details on the methods for collecting scats and identifying the frugivore species, see the original study[53] that involved *F. alnus*.

**Seedling recruitment.** To assess the probability of a seed of *F. alnus* to develop into a seedling, we conducted recruitment experiments during each year from 2016–2018. We collected fruits of at least six adults, removed the pulp, dried the seeds for 48 h at room temperature, and mixed them. At each of four study sites we established 10 marked plots, and at each of these plots we established subplots for the different years of the recruitment experiments, 2016 ($n = 40$ plots), 2017 ($n = 20$, only half the number of plots) and 2018 ($n = 40$). We sowed 25 seeds per subplot, thus, 2500 seeds in total. Each subplot had an area of 50 cm × 50 cm and was at least 5 m away from the nearest reproductive *F. alnus* individual. From 2017 to 2019, we checked the experimental subplots for the number and size of seedlings once per year in June and then tracked their fate as part of the demographic study. In addition, we checked once for emerging seedlings of *F. alnus* on a control subplot next to each subplot where we had not sown seeds. We found only one seedling of *F. alnus* in these control plots, indicating that external seed input into our recruitment plots was negligible. No seedlings were emerging in the year seeds were sown, confirming that seeds of *F. alnus* need cold-stratification before germination[49]. For further details on the methods for studying seedling recruitment, see the original study[15] that involved *F. alnus*.

**Plant vital rates.** To analyze the demography of *F. alnus*, we recorded the survival, growth, and reproduction of individuals of the tree from 2017–2019 at 14 study sites in the Białowieża Forest. We randomly chose individuals of all sizes in the study plots and attached tags to them to be able to relocate them throughout the study period. We measured the stem diameter of the tagged individuals of *F. alnus* at ground level with calipers or with a tape measure. We counted the number of fruits of adult trees before the main period of fruiting. However, in c. 27% of cases, we assessed a plant later in the season. Because animals had eaten some or even all of the fruits by that time, we only checked for leftovers of fruits to see if an individual was reproductive or not. Throughout this manuscript, individuals are categorized as reproductive if they had produced at least one fruit. Individuals that produced only flowers, but no seeds, were not considered reproductive because we were only interested in the outcome of reproduction (i.e., the production of propagules). We measured the diameter of 65 of the 253 first-year seedlings (26%). To include seedlings in the analysis whose diameters were not measured, we randomly assigned each of these seedlings a diameter based on the size distribution of the measured seedlings (mean = 0.767 cm, sd = 0.269 cm, lower bound = 0.380 cm). A large proportion of the plant individuals could not be relocated at each census because tags were often destroyed (c. 10–20%). Of the lost individuals, we were able to retrieve about 10% of individuals in the following study years. We recorded survival and death only for individuals which could be clearly identified throughout the study period. In total, we were able to record vital rates for 938 individuals, of which 341 were assessed once, 247 twice, and 350 for three consecutive years (following outlier detection, see statistical analyses).

**Canopy cover.** For studying effects of canopy cover on seed deposition, we split each of the 100 m transects into five 20 m segments. Every scat that was found along these transects was assigned to the closest segment. We took up to six hemispherical photos of the canopy with a fisheye lens at ground level along the transects in 2016 and 2017. To study effects of canopy cover on plant demography, we tagged most plants close to the transects (within 10 m distance), and assigned these individuals to the closest segment of the transects. If plant individuals were located farther away from the transects, we took up to three hemispherical photos within 10 m of those plants from 2016 to 2019. At the same times, we also took photos at the center of each of the 40 plots used to study plant recruitment. All photos were taken during the fruiting period of the fleshy-fruited plant community from June-October. The hemispherical photos were analyzed with DHPT 1.0[55] to calculate the proportion of area that was covered by the canopy at each location. By comparing the canopy cover derived from photos taken at the same location at different times, we found that canopy cover varied considerably over time resulting in a weak correlation between canopy cover values measured in 2016 and 2017 at the same locations (Supplementary Discussion 3, Supplementary Fig. 14). As a consequence, we did not use these data to model the between-year dynamics of the canopy of the forest[64]. Instead, we used local averages of the canopy cover (across seasons and years) as explanatory variables in the analyses of the seed deposition patterns and the vital rates.

**Estimating seed deposition patterns.** We collected 3632 scats with 15,382 seeds (during the 2016–2018 seasons) of the plant-frugivore community, of which 375 scats contained seeds of *F. alnus* (see also ref. [53]). Most scats containing seeds of *F. alnus* were from the birds *S. atricapilla* ($n = 232$), *T. merula* ($n = 55$) and *T. philomelos* ($n = 58$), and another 30 scats from the remaining animal community (Supplementary Table 3). The low sample size of scats with seeds of *F. alnus* allowed us only to predict seed deposition patterns of *F. alnus* for three of the 20 dispersers. To be able to include the remaining 17 animal species into the analyses, we made the following assumptions: (1) we pooled data on seed deposition of all animals across study sites and years in the forest assuming no temporal or spatial differences in our population. (2) We did not differentiate between seeds deposited beneath conspecific adults and elsewhere, as early recruitment of seedlings was not affected by conspecific adults (Supplementary Discussion 1). (3) We assumed that frugivores showed the same behavior and dropped scats at the same locations along the canopy gradient independent of whether they had eaten fruits of *F. alnus* or of one of the other 15 plant species fruiting at the same time. Thus, scats with seeds of species other than *F. alnus* were considered equally representative for deposition patterns of the different frugivores across the canopy gradient, and were used to study seed deposition pattern of *F. alnus*. We only pooled scats with seeds of other species that were found at the same time as scats with seeds of *F. alnus* in each year (except for *M. martes*, see next point). (4) We found scats of *M. martes* in the same micro-habitats (mostly the same tree logs) throughout the study years. This finding may be explained by the behavior of *M. martes* to use scats to mark home ranges and communicate with other individuals[107]. In order to have a sufficiently large sample size to analyze the deposition of seeds by *M. martes*, we pooled the scats containing seeds of fleshy-fruited plants from the entire fruiting period. (5) In our IPMs, we analyzed the role of rare frugivores separately, but treated them as functionally equal to each other. We pooled all data on fruit removal and seed deposition for frugivores with <10 scats (Supplementary Table 3) which may have masked subtle differences between contributing frugivore species. However, the number of samples was still too low to analyze seed deposition by rare frugivores ($n = 20$). To include rare frugivores in the analysis, we included scats

of animals that did not directly interact with *F. alnus*, but deposited seeds of other fleshy-fruited plants at sites where seeds of *F. alnus* were deposited. We assumed that the seed deposition by these animals, that are potentially able to interact with *F. alnus*, was equal to that of rare dispersers of *F. alnus* ($n = 16$ scats by six animal species, Supplementary Table 3). Only by analyzing the deposition of seeds of all species by rare dispersers were we able to estimate the contribution of these rare dispersers independently from that of frugivores that frequently interacted with *F. alnus*. We assumed that this approach will not qualitatively affect the results of the study, as differences in seed deposition by frugivores were not very pronounced (Fig. 1d). In addition, small differences in seed deposition by frugivore species did not affect seed dispersal quality as strongly as other components (e.g., seed predation; Fig. 3b).

However, over the course of 3 years we found >500 scats with seeds of *Sambucus nigra* beneath a single tree of that species, which produced over 40,000 fruits each year and attracted many frugivores. Seeds deposited beneath this *S. nigra* tree strongly affected the overall pattern of seed deposition of frugivores when we pooled the deposition data. We therefore used only scats containing seeds of *F. alnus* from the transect segment with the *S. nigra* tree in the analyses. Using this approach, we analyzed 1729 scat samples and differentiated between the seed deposition patterns of six animal species and one group of species collectively representing rare dispersers ('other species'). Each of the deposition patterns was based on a minimum number of 30 scat samples (Fig. 1c, Supplementary Table 3). We estimated seed deposition patterns for all animal species based on the scats with seeds from the entire plant community.

**Integral projection model.** To investigate the population dynamics of *F. alnus*, we used integral projection models (IPMs)[60,61]. We used stem diameter (log10-transformed and then standardized) at ground level as a state variable for size $z$ and the standardized, continuous canopy cover $c$ as a continuous state variable for the amount of shade. In the IPM, the individuals were thus characterized by two continuous state variables: their size and the degree of canopy cover at their location. The transition of the number of individuals $n$ of size $z$ in environment $c$ at time $t$ to the number of individuals $n'$ with size $z'$ in environment $c'$ at time $t + 1$ is given by

$$n'_{t+1}(z', c') = \left[ \iint_L^U P(z', z, c) + \sum_{x=i} \iint_L^U F(z', c', z, c) \right] n_t(z, c) dz dc \quad (1)$$

Here $P(z', z, c)$ describes survival, breakage, and growth of individuals as a function of their size and their environment, and $F(z', c', z, c)$ describes dispersal of seeds and recruitment of *F. alnus*. When seeds are taken up by animal $x$, the dispersed seed acquires a new location along the canopy gradient $c$. In contrast, when seeds are dispersed only by gravity or when animal $x$ drops the fruit beneath conspecific trees, the seeds will not be transferred to new locations and, thus, remain under the same canopy cover $c$ in which they were produced.

We used the 'cumulative kernel' (or 'bin-to-bin') approach to numerically integrate the IPM as it has been shown to perform better for slow-growing, long-lived species than the commonly used 'midpoint rule'[61,108]. In the integration $U$ was set to 1.1 times the upper and $L$ was set to 0.9 times the lower boundary of the observed size- and canopy-ranges within the forest. We added probabilities with smaller or larger values than the boundaries of the matrices to the outer classes to avoid eviction. We discretized the tree-IPM into $100 \times 100$ size classes with (standardized) sizes $z$ ranging from $-2.86$ to $2.36$, corresponding to a stem diameter of 0.06 cm and 89.68 cm, respectively. This resolution in size classes resulted in robust estimates of population growth rates ($\lambda$) as a quadrupling of the number of size classes hardly affected estimates of $\lambda$ (<0.001). The boundaries of the matrix did not correspond to the observed minimum (0.2 cm) and maximum diameter (18.6 cm; height = 9 m) of *F. alnus*. To keep the computational time for the IPMs within feasible limits, we split the canopy gradient into ten equally large segments. The (standardized) canopy cover ranged from $-4.279$ to $2.178$, corresponding to a range in canopy cover from 64.8% in bright environments to 92.1% in dark environments.

The transition of the population of *F. alnus* was based on the following equations:

$$n'_{t+1}(z', c') = \int_L^U S(z, c)(1 - B(z))G(z', z, c) n_t(z, c) dz dc \quad (2a)$$

$$+ \int_L^U S(z, c)B(z)R(z', z) n_t(z, c) dz dc \quad (2b)$$

$$+ \sum_{x=i} \iint_L^U f_{rel.anim.disp} f_{rel.int.freq}(x) f_{nocrush}(x) f_{consumed}(x) f_{seed} f_{deposition}(c', x) \begin{matrix} f_{repr}(z, c) f_{fruit}(z, c) \\ \\ f_{recruit} f_{dist}(z') n_t(z, c) dz dc \end{matrix} \quad (2c)$$

$$+ \sum_{x=i} \iint_L^U f_{rel.anim.disp} f_{rel.int.freq}(x) f_{nocrush}(x) 1 - f_{consumed}(x) f_{seed} \begin{matrix} f_{repr}(z, c) f_{fruit}(z, c) \\ \\ f_{recruit} f_{recfruit} f_{dist}(z') n_t(z, c) dz dc \end{matrix} \quad (2d)$$

$$+ \iint_L^U \left(1 - f_{rel.anim.disp}\right) f_{seed} \begin{matrix} f_{repr}(z, c) f_{fruit}(z, c) \\ \\ f_{recruit} f_{dist}(z') n_t(z, c) dz dc \end{matrix} \quad (2e)$$

In Eq. (2a), $S(z, c)$ models survival and $G(z', z, c)$ models growth to size $z'$ as a function of individual plant size $z$ and canopy $c$. However, growth of *F. alnus* is complex and some of the large individuals of *F. alnus* randomly broke from one year to the next. Most of these individuals died, which is covered by the survival function, but a few, usually large individuals ($n = 17$ over three years) survived breakage as they either had a small side-trunk before breakage or were able to re-sprout. The diameter of these broken individuals was strongly reduced at time $t + 1$. Thus, we split growth of *F. alnus* into two processes: Eq. (2a) describes growth of *F. alnus* for individuals that survived $S(z, c)$ and did not break with the probability $1 - B(z)$. Equation (2b) models the probability to survive $S(z, c)$ and break $B(z)$, and the size distribution $R(z', z)$ of broken individuals after re-sprouting. However, this event was very rare and, because of the low sample size, $B(z)$ and $R(z', z)$ were only a function of size and were kept constant among years and different canopies.

Our fecundity kernel for *F. alnus* included a model for consumed fruits (Eq. (2c)), a model for dropped fruits (Eq. (2d)), and a frugivore-independent gravity model (Eq. (2e)). Equation (2c), Eq. (2d), and Eq. (2e) are conditioned on each other. Each of these equations can be subdivided into three independent parts which take place chronologically in nature: fruit production, seed dispersal, and recruitment to the seedling stage. The first part describes the number of fruits produced by *F. alnus* and is given by the probability to reproduce $f_{repr}(z, c)$ and the number of fruits $f_{fruit}(z, c)$ as a function of size and canopy cover. The second part describes the process of dispersal of seeds by frugivore $x$ into canopy $c$. Here, $f_{rel.anim.disp}$ is the proportion of fruits dispersed by animals in the population. If we assume that all fruits are removed by frugivores, then $f_{rel.anim.disp} = 1$ applies and fecundity is only a function of Eq. (2c) and Eq. (2d). However, when the overall relative contribution of animal dispersal to fecundity decreases down to a point where no seeds are dispersed by animals ($f_{rel.anim.disp} = 0$), the gravity function (Eq. (2e)) becomes more important as ($1 - f_{rel.anim.disp}$) increases. Thus, $f_{rel.anim.disp}$ can be interpreted as the proportion of fruit removal by the animal community. The sum of Eq. (2c) and Eq. (2d) is the contribution of seed dispersal by a frugivore to population growth (see also Eq. (3)). $f_{rel.int.freq}(x)$ is the relative interaction frequency of frugivores with *F. alnus*. The interaction frequency was calculated as the product of the number of visits, the probability of handling a fruit during a visit, and the mean number of fruits that are handled when any fruits were handled. It describes the quantity component of seed dispersal effectiveness[3]. $f_{nocrush}(x)$ is the probability of a fruit not being crushed by frugivores $x$ and therefore not destroyed, $f_{consumed}(x)$ is the probability of a fruit being consumed conditional on not being crushed, $1 - f_{consumed}(x)$ is equivalent to the probability of a fruit being dropped beneath a conspecific adult, $f_{seed}$ is the mean number of seeds per fruit, $f_{deposition}(c', x)$ is the probability of a seed being deposited along the canopy gradient $c$ within the forest. The third part of Eq. (2c), Eq. (2d) and Eq. (2e) describes seedling recruitment; $f_{recruit}$ is the probability per seed to produce a seedling in the first year after dispersal, $f_{recfruit}$ is the factor by which $f_{recruit}$ is inhibited if seedlings are recruiting from seeds within a fruit ($-70\%$, see ref. [11]), i.e., when fruits are not eaten but fall or are dropped beneath parental trees, and $f_{dist}(z')$ is the size distribution of new seedlings at time $t + 1$.

Please note that we modeled the seed dispersal by all animal species that potentially disperse seeds (Supplementary Table 3). This means that we also included the seed dispersal by granivorous animals that predate most seeds (e.g., *Coccothraustes coccothraustes*). However, we corrected the effect of their seed dispersal for that of predation by including $f_{nocrush}$ in the model. The canopy $c$ in the definition of $f_{repr}$ and $f_{fruit}$ refers to the environment of the reproductive tree. As fruits dropped by frugivores (Eq. (2d)) or dispersed by gravity (Eq. (2e)) do not change their position along the canopy gradient, the canopy $c$ also refers to the environment of seeds of dropped or gravity-dispersed fruits (i.e., $c' = c$). In contrast, in Eq. (2c), the canopy $c'$ in the definition of $f_{deposition}$ refers to the new environment in which the seed is deposited after being taken up by frugivore $x$, and thus represents the effect of the transport of seeds from one environment to another.

**Calculating seed dispersal effectiveness.** The integral projection model ('IPM') was first used to calculate the SDE of gravity dispersal by calculating local population growth rates ($\lambda$) along the canopy gradient assuming that all individuals only occurred in a certain environment and that only gravity dispersal takes place. These local IPMs describe the growth of populations of *F. alnus* without dispersal of seeds between microhabitats (Fig. 1c). The local IPMs were then weighted by the relative abundance of the microhabitats along the canopy gradient in the forest, and summed to calculate the population growth rate ($\lambda$) without animal dispersal. To calculate the effect of animal seed dispersal on population growth rate, we gradually increased the importance of seed dispersal by animals in the IPMs (by increasing $f_{rel.anim.disp}$ from 0 to 1) and calculated the population growth rate. We also investigated the effects of animal seed dispersal during the colonization of favorable microhabitats ('gap colonization'). We assumed that *F. alnus* was not present in

forest gaps, defined as the 50% brightest environments along the canopy gradient in the forest. We calculated the IPM, but split the $P$ and $F$ kernels into two parts: (i) one part which modeled population growth along the entire canopy gradient as a function of animal dispersal, and (ii) another part which modeled population growth for dropped and gravity-dispersed fruits only in the 50% darkest environments. As we cut off the 50% brightest environments of the canopy gradient (i.e., 7.3% of all available microhabitats), we increased the relative abundance of the 50% darkest environments such that the values added up to 1. We are aware that this is a simple approach to investigate the effect of seed dispersal during colonization. At best, the effect of animal seed dispersal is modeled over time[31]. However, we were not able to track changes in the canopy structure of the forest over time (Supplementary Discussion 3), which prevented us from studying the effect of seed dispersal during forest succession[64].

The IPMs were further used to calculate the seed dispersal effectiveness of the different frugivore species[3]. Here, we defined the seed dispersal effectiveness of frugivores as the change in the population growth rate of *F. alnus* following the loss of a disperser species. This implied that, when a disperser was lost, the seeds were dispersed instead by gravity (i.e., interaction deficit). The proportion of seeds originally dispersed by the lost frugivore $x$ and dispersed by gravity instead was:

$$\iint_L^U \begin{matrix} f_{repr}(z,c)f_{fruit}(z,c) \\ f_{rel.anim.disp} f_{rel.int.freq}(x) f_{seed} \\ f_{recruit1} f_{recfruit} f_{dist}(z') n_t(z,c) dzdcdx \end{matrix} \qquad (3)$$

Then, we subtracted these frugivore-specific gravity components (Eq. (3)) from its frugivore-specific dispersal effects in the $F$ Kernel (Eq. (2c) and Eq. (2d)). In these IPMs, we made sure that all fruits were dispersed by the animals by setting $f_{rel.anim.disp}$ to 1. However, the relationship between population growth of *F. alnus* and the proportion of seeds dispersed by animals was non-linear and the slope of the curve decreased with increasing proportion of seeds dispersed (Fig. 2). A consequence of the shape of the curve is that plant populations with little dispersal will benefit more strongly from additional dispersal of their seeds by animals than those populations whose seeds are already dispersed to a large proportion.

To investigate the redundancy and complementarity of the seed dispersal by frugivores, we modeled the potential for interaction compensation by the remaining animal community following the loss of a frugivore species. To do so, we increased the relative interaction frequency of the remaining animal community by that of the lost dispersers, such that the summed proportion of removed fruits added up to 1 again. Because this step was computationally extensive, we modeled interaction compensation only for the four main dispersers.

The seed dispersal effectiveness of frugivore $x$ has both a quantity and quality component[3]: for the quantity component, we used the relative interaction frequency of a frugivore with *F. alnus*. For the quality component, we used the survival probability of a seed until the age of first reproduction as a measure when dispersed by a frugivore. The latter was calculated using a Markov chain, in which reproduction was an absorbing state in addition to mortality[61]. The survival probability of a seed of *F. alnus* until the age of first reproduction after being handled and dispersed by frugivore species $x$ is given by

$$f_{adulthood}(z_0, c', x)$$
$$= \iint_L^U f_{nocrush}(x) f_{consumed}(x) f_{deposition}(c', x) f_{recruit} l_{(\bar{a}_{repr})}(z_0, c') \, dz_0 dc \qquad (4a)$$

$$+ \iint_L^U f_{nocrush}(x) f_{drop}(c', x) f_{recruit} f_{recfruit} l_{(\bar{a}_{repr})}(z_0, c') dz_0 dc \qquad (4b)$$

which is the sum of the probability of a seed to produce a mature plant from consumed fruits (Eq. (4a)) and dropped fruits (Eq. (4b)). The probability of a seed to produce a mature plant was conditional on seed dispersal by frugivore $x$, the initial size of seedlings $z_0$ and the canopy cover $c'$. Here, $l_{(\bar{a}_{repr})}$ describes the probability of a seedling surviving until it has produced fruits at least once, and is a modification of formulas presented in ref. [61]. A more detailed derivation of $l_{(\bar{a}_{repr})}$ is given in Supplementary Methods 1.

### Statistical analyses

*Seedling recruitment.* We analyzed the effect of canopy cover and year on seedling recruitment in *F. alnus* with generalized linear mixed models with the number of seedlings that were recruited in spring and the number of non-recruited seeds in the same plot as a response variable. We included the identity of plots within sites as a random factor. In these models, we used a logit link and a beta-binomial error distribution to account for overdispersion.

*Plant vital rates.* To analyze the effects of study year, size of individuals, forest canopy cover, and their interaction on the vital rates of *F. alnus*, we used the study year, the standardized log10-transformed stem diameter and the standardized canopy cover as fixed factors and site as a random factor (see Supplementary Tables 4, 5). In addition, we added size$^2$ as a term in these models to test for non-linear relationships. However, many tags were lost in the field, making it difficult to relocate individuals of *F. alnus* over time. Often, thus, two or more different

individuals were incorrectly classified as the same individual, resulting in abnormal growth transitions (e.g., increases or decreases in diameter of >10 cm). To reduce the probability of including false data, we identified potential outliers using 2.24*standard deviation of the studentized residuals of the global model of plant growth as a threshold[109]. We removed 55 of the 1002 transitions of plant individuals from the dataset (c. 5.4% of total transitions): either entirely, if they had clearly erroneous values ($n = 14$), or by splitting records of single individuals into those of two or more independent individuals ($n = 41$).

To analyze the effect of year, size and canopy cover on the survival, breakage, and fruiting probabilities, we used a logit link and a binomial error distribution. In the analysis of the number of fruits, we used a log link, a Poisson error distribution and included an observation level random effect. To find the most parsimonious model for the analyses of survival, growth, fruiting probability, and the number of fruits, the global model and seven component models were ranked according to the small sample unbiased Akaike's information criterion (AICc, Supplementary Tables 4, 5, Supplementary Figs. 3, 4), using the R-package MuMIn version 1.43.17[110]. As the sample size was very low in the analyses of breakage and re-sprouting, we analyzed only the effects of size (linear) in the binomial models of breakage, and those of size and size$^2$ (linear and quadratic) in the growth analyses of re-sprouting individuals. In the analyses of re-sprouting, we further modeled the variance as a function of size and size$^2$ (Supplementary Table 4, Supplementary Fig. 8). We used the R-package glmmTMB version 1.1.2[58] and the R program version 4.1.1[59]. P-values were obtained with a Wald-$\chi^2$ test using the R-package car version 3.0-11[111]. The performance of all models was evaluated using the R-package DHARMa version 0.4.3[112].

*Seed dispersal effectiveness.* The change in population growth rate following the loss of a disperser species (i.e., the interaction deficit) was used as a measure for seed dispersal effectiveness[3]. To test if the seed dispersal effectiveness of animals was related to the quantity or quality of seed dispersal, we used Spearman rank correlations. However, we used partly the same parameters to calculate the quantity (fruit removal) and quality components (fruit handling behavior, seed deposition) for animal species (see Fig. 1, Supplementary Table 3). This resulted, for example, in equal contributions of animal species to population growth for which few data were available (*Apodemus flavicollis, Cervus elaphus, Dryomys nitedula*, unknown *Phasanidae, Prunella modularis, Sus scrofa*, see Fig. 3a). To avoid pseudo-replication, we calculated the correlations between SDE and the quantity and quality of seed dispersal with two sets of animal species. For the first set we used six dispersers separately and combined all others ($n = 7$, see Fig. 1c). For the second set we used 13 species separately and combined the rest ($n = 14$, see Fig. 3a). The correlations were qualitatively not affected in their sign or magnitude by the number of animal species differentiated or by the type of correlation analyses (Spearman vs. Pearson). We refer to the results of the Spearman rank correlations with $n = 14$ throughout the study.

**Quantifying uncertainty.** We used bootstrapping to examine the uncertainty in demography. We differentiated between uncertainty arising from observations of different plant individuals (hereafter 'growth uncertainty') and uncertainty arising from differences in the dispersal process (hereafter 'dispersal uncertainty'). To model uncertainty of growth, we resampled observations of individuals of *F. alnus* at each site in each year from the dataset, with replacement. To model uncertainty of dispersal, we resampled 500 times the removal observations, seed deposition, and seedling recruitment with replacement. The remaining parameters were kept constant. The number of replicates for each combination of year and site in the bootstrapped data was the same as in the field dataset. In addition, we made sure that every disperser species was present in each bootstrap sample. We calculated 500 IPMs using the bootstrap dataset and calculated the uncertainty as 95% prediction interval for each demographic process. The structure of the formulas of the IPM was held constant for the vital rates, i.e., we used the parameter estimates from the most parsimonious model (Supplementary Tables 4, 5) in all IPMs with the bootstrap data.

**Reporting summary.** Further information on research design is available in the Nature Portfolio Reporting Summary linked to this article.

## Data availability

The datasets generated during the current study are available in the Dryad Digital Repository, https://doi.org/10.5061/dryad.h44j0zpmq.

## Code availability

R code generated during the current study are available in the Dryad Digital Repository, https://doi.org/10.5061/dryad.h44j0zpmq.

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

## Acknowledgements

The authors acknowledge that they received funding from the German Research Foundation (FA925/10-1, FA925/10-2, BE 6041/1-1, SCHA 2085/1-2) in support for this research. We are deeply indebted to the numerous, enthusiastic volunteers who supported our work: Julius Angebauer, Ines Bischofberger, Leonie Braasch, Tessa Brandtner, Lea Dieminger, Levin Freitag, Amelie Hager, Elysia Hassen, Sabrina Hüpperling, Vincent Kramer, Wiebke Krug, Hanna Konrad, Tewannakit Mermagen, Nico Meyer, Simon Ostermann, Hannah Sunder-Plassmann, Frieder Thaler, Sophia Thiele, Zeynep Tür-kyilmaz, Martin Stankalla, Stella Weiß. We thank the administration of the Białowieża National Park, the forest administrations of Białowieża, Hajnówka and Browsk, and Polish authorities (Ministry of Environment, GDOS [Polish General Directorate of Environment Conservation, Warsaw] and RDOS [Regional Directorate of Environment Conservation, Białystok]) for the permission to carry out research in Białowieża Forest. We thank Esther Meißner and Marcel Becker for technical assistance. We thank Shripad Tuljapurkar and two anonymous reviewers for their constructive comments on earlier versions of the manuscript.

## Author contributions

F.R. contributed to conceptualization, methodology, investigation, formal analysis, visualization, writing the original draft, revising & editing; E.J. contributed to conceptualization, methodology, investigation, formal analysis, revising, supervision; J.S. contributed to conceptualization, methodology, writing review & editing; J.A. contributed to conceptualization, methodology, writing review & editing; H.F. contributed to methodology, writing review & editing; B.J. contributed to project administration, writing review & editing; D.M. contributed to investigation, writing review & editing; L.W. contributed to methodology, writing review & editing; N.F. contributed to conceptualization, investigation, writing review & editing, supervision, funding acquisition;

D.G.S. contributed to conceptualization, investigation, writing review & editing, supervision, funding acquisition.

## Funding

## Competing interests

The authors declare no competing interests.
