## [Peer Review File · Communications Biology]

Reviewers' comments:

Reviewer #1 (Remarks to the Author):

I like this paper -- important subject, nice complementary use of theory and data, lots of data (!), a great and relevant study system. But the writing needs improvement, as indicated by the comments below. Most importantly, you must define "efficiency (SDE)" and "interactions" at the start.

Line 101-102 "Only by leveraging data from over 10 years of seed dispersal research in the same ecosystem, we were able to seek ..." Rewrite: perhaps "Our 10 years of data on seed dispersal in one ecosystem allowed us to seek ..."

Line 115 Defines "interaction frequency" for the first time in this paper -- should be done at the beginning!

Line 123-126 The total is 18 but you started with 20 (on Line 115)?

Shripad Tuljapurkar

Line 214-5 Defines "SDE" for the first time in this paper -- should be done at the beginning!

Line 216 "We found an asymptotic pattern.." What does this mean? Asymptotic growth rate? Clarify.

Line 234-236 "... either of the following reasons: first, we found only small differences in seed deposition by animals.." I did not find a "second" anywhere!

Line 237 "Only large mammals partially avoid large tree logs when wolves are present, as these block escape routes" Either delete or explain the last phrase -- does not make sense to me.

Line 242 "we simulated virtually unrealistic patterns" Omit "virtually"

Reviewer #2 (Remarks to the Author):

The ms "Common seed dispersers contribute most to plant persistence" by Rehling et al explores the delayed outcome of frugivorous animal interactions with *Frangula alnus* on the demography of this plant species. The manuscript is very well written and enjoyable. The methods are extensive and well described and the results are relevant and contribute substantially to the field.

One of the strongest points that I find of this work, is the estimation of animal dispersal service for plants until dispersed seeds become a reproductive individual. Estimating the effectiveness of seed dispersers for such a late stage in the plant cycle is rare and highly valuable, as it gives a broad perspective into the long-term consequences of these animal-plant mutualisms. Authors also estimate the service that the animals, as a whole, provide to the plant population compared to a defaunation (no dispersers) scenario. These results are important for understanding population dynamics and the consequences of species loss on plant regeneration.

In my opinion, the manuscript title and conclusion rely rather too much on the finding that the quantity of interaction is a good surrogate of mutualism effectiveness. This result is not completely new, and has been previously reported in other studies that the authors cite, such as Vázquez et al 2005 or García-Rodríguez et al 2022. A smaller variation in the quality component of animals in

comparison with the wide variation of the quantity component accounts for why the latter is a better predictor for the final effectiveness. However, there is also evidence in other systems, with a functional and phylogenetic heterogeneous disperser assemblage, that the quality component can lead the variation and have stronger influence on the final effectiveness - e.g.:

- González-Castro, A., Calviño-Cancela, M. & Nogales, M. (2015). Comparing seed dispersal effectiveness by frugivores at the community level. *Ecology*, 96, 808–818.

- Brodie, J.F., Helmy, O.E., Brockelman, W.Y. & Maron, J.L. (2009). Functional Differences within a Guild of Tropical Mammalian Frugivores. *Ecology*, 90, 688–698. (cited in the manuscript).

Or evidence on complementary rather than redundant roles in seed deposition patterns for disperser species - e.g.:

- Bueno, R.S., Guevara, R., Ribeiro, M.C., Culot, L., Bufalo, F.S. & Galetti, M. (2013). Functional Redundancy and Complementarities of Seed Dispersal by the Last Neotropical Megafrugivores. *PLoS ONE*, 8, e56252.

It seems to me that the main conclusion is too generalised coming from a single species study. It is possible that a stronger effect of quantity in the dispersal effectiveness happens in systems where the main dispersers have similar roles in their dispersal quality, as opposed to other more heterogeneous systems (e.g. oceanic) whose main dispersers present complementary roles.

To me, the highest value of this manuscript rests on the successful quantification of the dispersal service by animals up to such advanced stages of the plant cycle, having the differential contribution to population growth disaggregated for the main dispersers and compared to gravity dispersal. The study builds into previous work by providing high quality long-term data and by incorporating a more refined and advanced quality component. I suggest authors give more weight to these results and de-emphasize or relativize the importance of the quantity component to their study system.

The literature cited is well chosen and exhaustive, providing a very good state of art of the central topic of the manuscript and a rich discussion.

Finally, I have read the models section thoroughly, and it is my impression that they are sufficiently detailed and that the assumptions made by the authors are justified. I present some minor comments regarding analysis that have escaped my understanding. However, I feel compelled to point out that I have never worked with IPM models and I lack the confidence to review them in depth, so hopefully another reviewer can examine this aspect of the manuscript more thoroughly.

Below I present other minor comments referring line numbers:

Line 91: I suggest referencing to Fig.1 in general (with no letter) as Fig.1d also presents the canopy cover gradient effect indicated in the phrase.

Line 98-99: a short add on to this phrase indicating that seedling recruitment was measured through sowing experiments in the field would help clarifying the methodology for this specific step.

Line 104: "Is the seed dispersal of plants by frugivorous animals a mutualism?". This question's answer seems evident and widely supported in literature, also not specifically targeted in the study or backed up by the results, i.e. authors are not looking at both partner's benefits during the interaction.

Line 105-106: The motivation to explore this question (only the quantitative component) is not previously justified in the introduction. For a first time reader this comes as a surprise (i.e. why compensation cannot be qualitative?). I imagine this question is based on later results where quality

turns out to be highly similar between dispersers, but this piece of information is presented later on.

Line 110-112: An important consideration in this phrase is that species redundancy refers to their *quality* of dispersal. For one species to replace another, the new species must reach the same level of consumption as the previous one, and this assumption may be constrained by disperser abundances or dietary preferences - as authors also discuss in lines 310-320.

Line 158: reference to Fig.1 is unnecessary

Line 175-176: weird phrasing

Line 194: in figure 2 the λ does not seem to reach 1.00 at 100% dispersal, as indicated in the text, could there be some kind of figure distortion?

Line 250-254: open area effects can be highly variable (not necessarily negligible), even in arid ecosystems, e.g. Calviño-Cancela, M. & Martín-Herrero, J. (2009). Effectiveness of a varied assemblage of seed dispersers of a fleshy-fruited plant. *Ecology*, 90, 3503–3515.

Line 294: replace 'when' by 'if'?

Line 463: if the individual was only recorded once how is it possible to use this information as a transition probability in the model?

Line 474: I am confused about the number of photos taken per transect. In the main text it says up to 6 between years 2016-19, but then in Suppl. Discussion 3 the first figure shows 5 periods for years 2016-17. Why was the correlation between years only performed for 2016 and 2017?

Line 553: 'scats from the entire plant community' sounds weird - I suggest clarifying the scats contain seeds from the entire plant community

Line 615: was this inhibition factor for dropped seeds in fruit (i.e. not depulped?) taken from Rogers et al 2021? If it is just one value, it would be informative to provide it in the text.

Line 629-631: these two phrases one after the other are confusing. Shouldn't be 'first' only the gravity model?

Line 649-651: it is unclear to me why the 50% darkest environments relative abundance needed to be increased and the lightest decreased for the gap-dependent model.

Line 745: which are the parameters in common between the quantity and quality component?

Please check the caption in figure number 2, it is not clearly explained the difference between the two dashed lines, also the prediction intervals indicated in the caption cannot be seen in the figure.

Does the vertical black line in Fig 1c represent the median of the distribution? please indicate its meaning in the caption.

Reviewer #3 (Remarks to the Author):

The manuscript entitled "Common seed dispersers contribute most to plant persistence" focused on the evaluation of the effect of different seed disperser species on the persistence of a plant species.

Authors collected, used and analyzed a variety of data related with the seed dispersal process and population dynamics of *Frangula alnus*; this is a strong point of the manuscript. Moreover, the evaluation of long-term effects of animal seed dispersal for plant populations growth reflects the ecological importance of this process.

I would suggest including at some part of the manuscript that the results found in this work depend on the dependence of the plant population growth to seed dispersal by animals. To other plants, the seed dispersal stage may have a different effect on their population's growth. Some other variables can be strong at regulating different stages in the plant life cycle.

My specific comments to the manuscript are listed below.

Abstract

L26: here you stated that "animal dispersal increased population growth by 2.5%". I would recommend indicating the time (or number of plant generations) in which you analyzed population growth.

L29-31: I think, in this part of the sentence, you are generalizing the results found here and maybe in other study cases you may obtain different results. I would suggest expressing conclusions for this work with caution.

Introduction

L103-104: I think that studying just one case of a plant species dispersed by animals it is not enough for answering the second question. I would strongly recommend focusing these questions to this study case.

L108: what authors mean with the expression "animals outperforms seed dispersal by gravity"? what kind of seed dispersal are you referring to? Please explain it or modify the sentence.

L110: what are you referring to "This indicates that seed dispersing animals are redundant..."? Previous sentence refers to interaction frequency. Please state the indicators you used to determine that "animal species are redundant as seed dispersers".

Results and Discussion

L203: Please clarify "animal seed dispersal in plants of this species..."

L248: in what other type of habitats? please specify this.

L298: had you find any difference about birds and mammals in their ability to compensate seed dispersal functions?

L331: you used the term "common" throughout the manuscript, also in its title. However, I believe it will be important to clarify what you considered as a "common seed disperser", is this related with its abundance or frequency of interactions? Please include this aspect in the text.

Methodology

L378: have you applied the same methodology for surveying birds and mammals? indicate this in text.

Please note: In the point-by-point response to reviewers, our **answers to comments are given in red**, *quotations from the manuscript in italic*, line numbers refer to the revised manuscript without changes tracked).

Reviewers' comments:

Reviewer #1 (Remarks to the Author):

I like this paper -- important subject, nice complementary use of theory and data, lots of data (!), a great and relevant study system. But the writing need improvement, as indicated by the comments below. Most importantly, you must define "efficiency (SDE?" and "interactions" at the start.

Many thanks for your review and the positive feedback. We appreciate your interest in our manuscript and your valuable comments. We thoroughly revised the entire manuscript text. We included definitions of SDE (line 72-73: *"The overall effect of a certain seed-dispersing animal species on a plant population has been called seed dispersal effectiveness (SDE)."*) and interaction frequency (see below) in the introduction. We hope that our revision improved the ms and fully addressed your concerns.

Line 101-102 "Only by leveraging data from over 10 years of seed dispersal research in the same ecosystem, we were able to seek ..." Rewrite: perhaps "Our 10 years of data on seed dispersal in one ecosystem allowed us to seek ..."

Changed as suggested.

Line 115 Defines "interaction frequency" for the first time in this paper -- should be done at the beginning!

We now define the quantity of seed dispersal and quality of seed dispersal separately in the introduction by stating:

"The quantity of seed dispersal is equal to the frequency of interactions between animals and plants. The interaction frequency of seed dispersal can be expressed as the number of visits to a plant by an animal multiplied by the number of seeds dispersed per visit." (line 75-78)

Line 123-126 The total is 18 but you started with 20 (on Line 115)?

Yes, indeed, you are right. The total was 20. We now state:

"Four bird species accounted for 86.6% (95% confidence interval: 82.4-89.8%) of the interactions: Sylvia atricapilla (58.7%, 53.2-63.9%), Turdus merula (15.0%, 9.8-20.6%), Erithacus rubecula (8.3%, 6.2-10.4%) and T. philomelos (4.6%, 2.9-6.5%), while each of the sixteen remaining animal species contributed ~1% or less to total fruit removal (Supplementary Table 3, Supplementary Figure 2)." (line 133-137)

Line 214-5 Defines "SDE" for the first time in this paper -- should be done at the beginning!

We previously had defined seed dispersal effectiveness, but not its abbreviation in the introduction. We now included the abbreviation in the introduction. We hope this clarifies the term ‘total seed dispersal effectiveness’. We now state:

“The overall effect of a certain seed-dispersing animal species on a plant population has been called seed dispersal effectiveness (SDE)³. The probably most comprehensive measure of SDE is the effect of seed dispersal on the population growth rate of a plant. The SDE can be quantified as the product of the quantity and the quality of seed dispersal. The quantity of seed dispersal is equal to the frequency of interactions between animals and plants. The interaction frequency of seed dispersers can be expressed as the number of visits to a plant by an animal, multiplied by the number of seeds dispersed per visit. The quality of seed dispersal describes the effect of dispersal on the fate of a seed until it develops into an adult plant.” (line 72-79)

Line 216 "We found an asymptotic pattern.." What does this mean? Asymptotic growth rate? Clarify.

In the old manuscript version, the “asymptotic pattern” described the shape of the curve of how the loss of seed dispersers will affect the asymptotic population growth rate λ . We agree with you that this framing was confusing and deleted it. We have rephrased the description as follows:

*“We found that the loss of seed dispersal by some animal species reduced the population growth rate of *F. alnus* (Fig. 3a): The reduction was strongest when either *S. atricapilla* (- 47.2%), *T. merula* (-10.1%), *E. rubecula* (- 5.5%) or *T. philomelos* (- 3.0%) were lost. In contrast, the extinction of each of the 14 other animal species reduced the population growth of *F. alnus* by less than 1% (representing together 33% of the entire studied frugivore community and 70% of the dispersers of *F. alnus*). “ (line 226-231)*

Line 234-236 " ... either of the following reasons: first, we found only small differences in seed deposition by animals.." I did not find a "second" anywhere!

To improve readability and clarity, we now state the five possible reasons at the beginning and number them. We then expand our arguments referring to the number.

*“The strong relationship between SDE and the interaction frequency of animals with *F. alnus* may have been due to the following not mutually exclusive reasons: (1) small differences among species in patterns of seed deposition, (2) small effects of canopy cover on seedling recruitment, (3) small effects of density dependence on seedling recruitment, (4) low predation of seeds, (5) the assumption that passage through the gut of all dispersers had the same effect on seed germination.” (line 242-247)*

Line 237 "Only large mammals partially avoid large tree logs when wolves are present, as these block escape routes" Either delete or explain the last phrase -- does not make sense to me.

Thank you for pointing out this uncertain unclear phrase. In Białowieża forest, wolves are common predators. As a consequence, large mammalian herbivores avoid browsing near large tree logs because wolves may hide behind them or because they block escape routes when wolves are approaching the mammals. As a consequence,

large tree logs may influence the fine-scale seed deposition of large mammals in the forest. However, large mammals only infrequently fed on fruits of *F. alnus* (< 1%), and pointing out this exception may not be very relevant. We therefore removed this sentence.

Line 242 "we simulated virtually unrealistic patterns" Omit "virtually"

Changed as suggested.

Shripad Tuljapurkar

Reviewer #2 (Remarks to the Author):

The ms “Common seed dispersers contribute most to plant persistence” by Rehling et al explores the delayed outcome of frugivorous animal interactions with *Frangula alnus* on the demography of this plant species. The manuscript is very well written and enjoyable. The methods are extensive and well described and the results are relevant and contribute substantially to the field.

Many thanks for your very thorough review, your positive evaluation and many valuable suggestions, which helped us to improve our manuscript. We hope that we have appropriately addressed your remarks.

One of the strongest point that I find of this work, is the estimation of animal dispersal service for plant up until dispersed seeds become a reproductive individual. Estimating the effectiveness of seed dispersers for such a late stage in plant cycle is rare and highly valuable, as it gives a broad perspective into the long term consequences of these animal-plant mutualisms. Authors also estimate the service that the animals, as a whole, provide to the plant population compared to a defaunation (no dispersers) scenario. These results are important for understanding population dynamics and the consequences of species loss on plant regeneration.

Thank you for your appreciation of our manuscript.

In my opinion, the manuscript title and conclusion relies rather too much on the finding that the quantity of interaction is a good surrogate of mutualism effectiveness. This result is not completely new, and has been previously reported in other studies that the authors cite, such as Vázquez et al 2005 or García-Rodríguez et al 2022. A smaller variation in the quality component of animals in comparison with the wide variation of the quantity component accounts for why the latter is a better predictor for the final effectiveness. However, there is also evidence in other systems, with a functional and phylogenetic heterogenous disperser assemblage, that the quality component can lead the variation and have stronger influence on the final effectiveness - e.g.:

- González-Castro, A., Calviño-Cancela, M. & Nogales, M. (2015). Comparing seed dispersal effectiveness by frugivores at the community level. *Ecology*, 96, 808–818.

- Brodie, J.F., Helmy, O.E., Brockelman, W.Y. & Maron, J.L. (2009). Functional Differences within a Guild of Tropical Mammalian Frugivores. *Ecology*, 90, 688–698. (cited in the manuscript).

Or evidence on complementary rather than redundant roles in seed deposition patterns for disperser species - e.g.:

- Bueno, R.S., Guevara, R., Ribeiro, M.C., Culot, L., Bufalo, F.S. & Galetti, M. (2013). Functional Redundancy and Complementarities of Seed Dispersal by the Last Neotropical Megafrugivores. *PLoS ONE*, 8, e56252.

It seems to me that the main conclusion is too generalised coming from a single species study. It is possible that a stronger effect of quantity in the dispersal effectiveness happens in

systems where the main dispersers have similar roles in their dispersal quality, as opposed to other more heterogenous systems (e.g. oceanic) whose main dispersers present complementary roles.

Thank you for your extensive thoughts on our main conclusion. The reviewer raises several points which we will address separately below: (1) quantity vs. quality of seed dispersal for determining SDE, (2) the redundancy and complementarity of provided services, and (3) the generalisation based on our study system, while indicating how we changed the manuscript accordingly.

- (1) SDE: We agree with the reviewer that it has been suggested before that interaction frequency might be a suitable surrogate for the effectiveness of animal-plant interaction. However, at best, these studies have examined the effects of animals on the early stages of regeneration (see e.g. González-Castro et al. 2015, 2021; García-Rodríguez et al. 2022). For example, the highly influential study of Vazquez et al. (2005) used the number of seeds/fruits removed per visit to a plant as a measure of the quality of seed dispersal. Not tracking the effects of animal dispersal through the entire life cycle of a plant neglects that processes acting at late life cycle stages are independent of the frugivores and environment- and context-dependent (Schupp et al. 2007, Godínez-Alvarez & Jordano 2007). As integrating post-dispersal stages of the plant life cycle into quality of dispersal is expected to have profound effects on SDE, numerous authors have pointed out the need to fill this research gap in seed dispersal ecology (Wang & Smith 2002, Godínez-Alvarez & Jordano 2007, Schupp et al. 2010, Côrtes & Uriarte 2013, Schupp et al. 2017, Rogers et al. 2019, Beckman et al. 2020, Rogers et al. 2021).

Some of the studies the reviewer mentioned, e.g. Brodie et al. (2009), are notable exceptions, as they were able to link animal seed dispersal to population growth of a plant. However, these studies only investigated up to a maximum of five animal dispersers, making it impossible to study SDE-quantity/quality relationships at the disperser community level.

We discuss all those studies in our manuscript, but also emphasize that our finding that interaction frequency is a good surrogate of seed dispersal effectiveness across 20 animal species. This is in our opinion one of the key findings with broad implications for the fields of ecology, evolution and nature conservation (see also Kleijn et al. 2015, Winfree et al. 2015, Winfree 2020). Yet, we agree with the reviewer that we have certainly overgeneralized the results of our study, and have now toned down our conclusions (see point 3 below).

- (2) Functional redundancy: We agree with the reviewer that dietary preferences of animals will be important for whether interactions of lost seed dispersers will or will not be compensated by other animal species. We discussed this limitation in the manuscript (see line 322-335).

With respect to the study mentioned by the reviewer: our study and that of Bueno et al. (2013) investigated different aspects of SDE (local population growth vs. regional spread of populations/gene flow between populations), making it difficult to compare patterns in functional redundancy/complementarity. We were aware of this aspect, and discuss the potential functional redundancy of animal dispersers only with respect to the local population growth of *F. alnus*, and not with respect to gene flow or

population spread. However, to clarify that the functional redundancy of animal dispersers will also depend on other aspects of population growth, we now state in the text:

“Moreover, the loss of certain animals could affect population processes that are beyond the scope of this study, for instance range expansion^{32,88}, gene flow⁸⁹ and plant migration in response to climate change^{17,18}” (line 332-335)

- (3) We agree with reviewer 2 and 3 that we have potentially overgeneralized the findings of our study, as we only studied one plant population in one ecosystem. To address this issue, we changed the title of the manuscript, reduced the overgeneralization and emphasized that the conclusions are drawn from the results of a study on only one plant population. We changed the title of the manuscript to

“Common seed dispersers contribute most to the persistence of a fleshy-fruited tree” (line 1);

We toned down the conclusion section in the abstract and changed it to:

“Our results support the notion that frequently interacting mutualists contribute most to the persistence of the populations of their partners, [...]” (line 29-31)

We also changed the concluding paragraph of the introduction:

“By analyzing the whole life cycle of a fleshy-fruited tree, this study shows that its population growth is increased by the seed dispersal by frugivorous animals compared to seed dispersal by gravity, especially when trees need to colonize forest gaps. The interaction frequency of seed-dispersing animals was found to be a suitable predictor for the total effect of the animals on plant demography. Our results support the notion that seed-dispersing animal species in temperate forests may be largely redundant in the quality of the services they provide. [...]” (line 112-117);

Moreover, we changed parts of the conclusions:

*“Projected over the life cycle of *F. alnus*, seed dispersal by animals has positive effects on its population growth.”* (line 348-349);

and

“We further conclude that differences in interaction frequency between mutualists may typically be more pronounced than differences in the effects of interaction quality on partners. Therefore, the interaction frequency may be a suitable surrogate for the total effect of mutualists on their partners⁴.” (line 353-356)

To me, the highest value of this manuscript rests on the successful quantification of the dispersal service by animals up to such advanced stages of the plant cycle, having the differential contribution to population growth disaggregated for the main dispersers and compared to gravity dispersal. The study builds into previous work by providing high quality

long-term data and by incorporating a more refined and advanced quality component. I suggest authors give more weight to these results and de-emphasize or relativize the importance of the quantity component to their study system.

We agree with the reviewer that the effects of seed dispersal by frugivores on plant populations is a key feature of our study. While we believe that the positive effects of frugivores on the plant population were sufficiently highlighted in the previous version of the abstract and the main text, we now also emphasize them in the conclusion:

“Projected over the full life cycle of F. alnus, seed dispersal by animals has positive effects on its population growth. The seed dispersal mutualism is relatively robust to the loss of single disperser species, because most animal species only consume few fruits which makes them functionally less relevant.” (line 348-351)

The literature cited is well chosen and exhaustive, providing a very good state of art of the central topic of the manuscript and a rich discussion.

Thank you.

Finally, I have read the models section thoroughly, and it is my impression that they are sufficiently detailed and that the assumptions made by the authors are justified. I present some minor comments regarding analysis that have escaped my understanding. However, I feel compelled to point out that I have never worked with IPM models and I lack the confidence to review them in depth, so hopefully another reviewer can examine this aspect of the manuscript more thoroughly.

Below I present other minor comments referring line numbers:

Line 91: I suggest referencing to Fig.1 in general (with no letter) as Fig.1d also presents the canopy cover gradient effect indicated in the phrase.

Changed as suggested.

Line 98-99: a short add on to this phrase indicating that seedling recruitment was measured through sowing experiments in the field would help clarifying the methodology for this specific step.

Thank you for this suggestion. We now state:

“In a fourth step, we investigated seedling recruitment by experimentally sowing 2,500 seeds¹⁵.” (line 103-104)

Line 104: “Is the seed dispersal of plants by frugivorous animals a mutualism?”. This question’s answer seems evident and widely supported in literature, also not specifically targeted in the study or backed up by the results, i.e. authors are not looking at both partner’s benefits during the interaction.

We have removed the second question.

Line 105-106: The motivation to explore this question (only the quantitative component) is not previously justified in the introduction. For a first time reader this comes as a surprise (i.e.

why compensation cannot be qualitative?). I imagine this question is based on later results where quality turns out to be highly similar between dispersers, but this piece of information is presented later on.

We now state:

“(iii) Can the interactions of extinct species be functionally compensated by the remaining animal community?” (line 109-111)

Line 110-112: An important consideration in this phrase is that species redundancy refers to their ***quality*** of dispersal. For one species to replace another, the new species must reach the same level of consumption as the previous one, and this assumption may be constrained by disperser abundances or dietary preferences - as authors also discuss in lines 310-320.

We now state:

“Our results support the notion that seed-dispersing animal species in temperate forests may be largely redundant in the quality of the services they provide. Seed dispersers could potentially take over the ecological role of other species if they quantitatively compensate lost interactions.” (line 116-119)

Line 158: reference to Fig.1 is unnecessary

Removed as suggested.

Line 175-176: weird phrasing

We now state:

*“We first quantified the effect of local canopy cover on population growth of *F. alnus* (Fig. 1d). For this, we modelled the population growth of *F. alnus* at several points along the canopy gradient and assumed that there was only gravity dispersal and all seeds would thus be deposited in the same environment as their parents.” (line 182-185)*

Line 194: in figure 2 the λ does not seem to reach 1.00 at 100% dispersal, as indicated in the text, could there be some kind of figure distortion?

This difference arises from rounding. Lambda was 0.9966 at 100% dispersal.

Line 250-254: open area effects can be highly variable (not necessarily negligible), even in arid ecosystems, e.g. Calviño-Cancela, M. & Martín-Herrero, J. (2009). Effectiveness of a varied assemblage of seed dispersers of a fleshy-fruited plant. *Ecology*, 90, 3503–3515.

We now included that study (citation 67) in the previous sentence:

“For example, in arid environments such as deserts, most seedlings survive only in the shade of nurse plants where they are protected from heat stress⁶⁵, while seeds dispersed by gravity or deposited by animals in open areas do not contribute to plant regeneration²⁷ (but see^{66,67}).“ (line 263-266)

Line 294: replace ‘when’ by ‘if’?

Changed as suggested.

Line 463: if the individual was only recorded once how is it possible to use this information as a transition probability in the model?

Individuals that were measured once were not used in the analyses of plant survival and growth. Instead, these individuals were only used to relate reproduction (i.e. fruiting probability and number of fruits) to plant size.

Line 474: I am confused about the number of photos taken per transect. In the main text it says up to 6 between years 2016-19, but then in Suppl. Discussion 3 the first figure shows 5 periods for years 2016-17. Why was the correlation between years only performed for 2016 and 2017?

We took hemispherical photos along the deposition transects only in 2016 and 2017, and at a minority of plots from 2016 to 2019 (21% of total sample). We were thus only able to study the correlation between the canopy cover in 2016 and 2017. To avoid confusion, we restructured the text and now state:

“For studying effects of canopy cover on seed deposition, we split each of the 100 m transects into five 20 m segments. Every scat that was found along these transects was assigned to the closest segment. We took up to six hemispherical photos of the canopy with a fisheye lens at ground level along the transects in 2016 and 2017. To study effects of canopy cover on plant demography, we tagged most plants close to the transects (within 10 m distance), and assigned these individuals to the closest segment of the transects. If plant individuals were located farther away from the transects, we took up to three hemispherical photos within 10 m of those plants from 2016 to 2019. At the same times we also took photos at the center of each of the 40 plots used to study plant recruitment.” (line 489-497)

Line 553: ‘scats from the entire plant community’ sounds weird - I suggest clarifying the scats contain seeds from the entire plant community

Changed as suggested.

Line 615: was this inhibition factor for dropped seeds in fruit (i.e. not depulped?) taken from Rogers et al 2021? If it is just one value, it would be informative to provide it in the text.

Yes, it was taken from Rogers et al 2021. We now also cite this value in the MM section as suggested.

Line 629-631: these two phrases one after the other are confusing. Shouldn't be ‘first’ only the gravity model?

We now state:

*“The integral projection model (‘IPM’) was first used to calculate the SDE of gravity dispersal by calculating local population growth rates (λ) along the canopy gradient assuming that all individuals only occurred in a certain environment and that only gravity dispersal takes place. These local IPMs describe the growth of populations of *F. alnus* without dispersal of seeds between microhabitats (Fig. 1c). The local IPMs were then weighted by the relative abundance of the microhabitats along the canopy gradient in the forest, and summed to calculate the population growth rate (λ) without animal dispersal.” (line 652-658)*

Line 649-651: it is unclear to me why the 50% darkest environments relative abundance needed to be increased and the lightest decreased for the gap-dependent model.

Here, we modelled gravity dispersal. We assumed that, then, no *F. alnus* plants are growing in forest gaps, and thus, there is no seed dispersal of *F. alnus* to these forest gaps. However, cutting off half the canopy gradients implies that the proportions of area of the remaining microhabitats do not add up to 1 anymore, which is why we increased the relative abundance of the 50% darkest environments. Not increasing the relative abundance of microhabitats would have led to an unintentional loss of seeds in the IPM, and to a decrease of population growth.

To make this clearer, we now state:

“As we cut off the 50% brightest environments of the canopy gradient (i.e. 7.3% of all available microhabitats), we increased the relative abundance of the 50% darkest environments such that the values added up to 1.” (line 667-669)

Line 745: which are the parameters in common between the quantity and quality component?

Whether we used the same parameters for the quantity or quality components depended on the plant life cycle stage. For example, some animals were only identified as disperser of *F. alnus* by collecting their scats (Schlautmann et al. 2021). As we did not observe the fruit removal and the fruit handling behaviour of these animal species, we modelled their behaviour similar to that of rare dispersers of *F. alnus* which were observed during seed removal (Albrecht et al. 2013). Similarly, we could only differentiate seven deposition patterns, of which one deposition pattern (‘Other’) represented the deposition patterns of all rare dispersers combined. We have clarified this part of the text by stating:

“However, we used partly the same parameters to calculate the quantity (fruit removal) and quality components (fruit handling behavior, seed deposition) for animal species (see Fig 1, Supplementary Table 3). This resulted, for example, in equal contributions of animal species to population growth for which few data were available (Apodemus flavicollis, Cervus elaphus, Dryomys nitedula, unknown Phasianidae, Prunella modularis, Sus scrofa, see Fig. 3a). To avoid pseudo-replication, we calculated the correlations between SDE and the quantity and quality of seed dispersal with two sets of animal species. For the first set, we used six dispersers separately and combined all others ($n = 7$, see Fig. 1c). For the second set we used 13 species separately and combined the rest ($n = 14$, see Fig. 3a). The correlations were qualitatively not affected in their sign or magnitude by the number of animal species differentiated or by the type of correlation analyses (Spearman vs. Pearson). We refer to the results of the Spearman rank correlations with $n = 14$ throughout the study.” (line 756-767)

Please check the caption in figure number 2, it is not clearly explained the difference between the two dashed lines, also the prediction intervals indicated in the caption cannot be seen in the figure.

We changed the figure caption, and added two more sentences for clarification. Please also note that the prediction intervals are illustrated, but they are extremely small. We have changed the figure caption to:

“The effect of seed dispersal by 20 animal species on the population growth rate of F. alnus in Białowieża Forest, Eastern Poland. Two scenarios are presented: (1) Plant individuals occur along the entire canopy gradient (‘fully established’, short-dashed) or (2) occur only in closed forest (the 50%-darkest microhabitats, i.e. 92.7% of available area) and depend therefore on animal seed dispersal for establishment in forest gaps (‘gap colonization’, long-dashed line). In a population ‘fully established’ occurring along the full canopy gradient, gravity dispersal and fruits dropped by animals can result in successful plant regeneration where the canopy is not closed. In comparison, the population growth rate is reduced in a ‘gap colonization’ population if gravity dispersal is predominant, because plant establishment and growth is much reduced under a closed canopy. When 100% of seeds are dispersed by animals, the long-term stable stage distribution of plants and their distribution along the canopy cover is identical, and so is the population growth rate λ . Prediction intervals (95%) based on 500 non-parametric bootstraps of dispersal data were calculated for the effect of the proportion of seeds dispersed on λ . Please note that uncertainty in the effect of seed dispersal resulted in only extremely small differences in the population growth rate of F. alnus.” (line 1095-1109)

Does the vertical black line in Fig 1c represent the median of the distribution? please indicate its meaning in the caption.

Yes, indeed. We now included:

“A vertical line indicates the median of a distribution.” (line 1090)

Reviewer #3 (Remarks to the Author):

The manuscript entitled “Common seed dispersers contribute most to plant persistence” focused on the evaluation of the effect of different seed disperser species on the persistence of a plant species. Authors collected, used and analyzed a variety of data related with the seed dispersal process and population dynamics of *Frangula alnus*; this is a strong point of the manuscript. Moreover, the evaluation of long-term effects of animal seed dispersal for plant populations growth reflects the ecological importance of this process.

I would suggest including at some part of the manuscript that the results found in this work depend on the dependence of the plant population growth to seed dispersal by animals. To other plants, the seed dispersal stage may have a different effect on their population’s growth. Some other variables can be strong at regulating different stages in the plant life cycle.

Many thanks for your review and the positive feedback. We really appreciate your interest in our manuscript. In response to your feedback and the feedback by reviewer 2 and the editor, we reduced the overgeneralization of our conclusions throughout the text. We changed the title of the manuscript to

“Common seed dispersers contribute most to the persistence of a fleshy-fruited tree” (line 1);

We toned down the conclusion section in the abstract and changed it to:

“Our results support the notion that frequently interacting mutualists contribute most to the persistence of the populations of their partners, [...]” (line 29-31)

We also changed the concluding paragraph of the introduction:

“By analyzing the whole life cycle of a fleshy-fruited tree, this study shows that its population growth is increased by the seed dispersal by frugivorous animals compared to seed dispersal by gravity, especially when trees need to colonize forest gaps. The interaction frequency of seed-dispersing animals was found to be a suitable predictor for the total effect of the animals on plant demography. Our results support the notion that seed-dispersing animal species in temperate forests may be largely redundant in the quality of the services they provide. [...]” (line 112-117);

Moreover, we changed parts of the conclusions:

*“Projected over the life cycle of *F. alnus*, seed dispersal by animals has positive effects on its population growth.”* (line 348-349);

and

“We further conclude that differences in interaction frequency between mutualists may typically be more pronounced than differences in the effects of interaction quality on partners. Therefore, the interaction frequency may be a suitable surrogate for the total effect of mutualists on their partners⁴.” (line 353-356)

My specific comments to the manuscript are listed below.

Abstract

L26: here you stated that "animal dispersal increased population growth by 2.5%". I would recommend indicating the time (or number of plant generations) in which you analyzed population growth.

The demographic model investigated how the finite rate of population growth (λ) would be if the conditions in the future were the same as during the study period (once a steady state has been reached). Thus, a specific time cannot be given.

L29-31: I think, in this part of the sentence, you are generalizing the results found here and maybe in other study cases you may obtain different results. I would suggest expressing conclusions for this work with caution.

We now state more cautiously:

"Our results support the notion that frequently interacting mutualists contribute most to the persistence of the populations of their partners, [...]" (line 29-31)

Introduction

L103-104: I think that studying just one case of a plant species dispersed by animals it is not enough for answering the second question. I would strongly recommend focusing these questions to this study case.

Reviewer 2 similarly criticized this question. We decided to remove it.

L108: what authors mean with the expression "animals outperforms seed dispersal by gravity"? what kind of seed dispersal are you referring to? Please explain it or modify the sentence.

Gravity dispersal means that seeds simply fall to the ground and are not dispersed any further. We now state:

"By analyzing the whole life cycle of a fleshy-fruited tree, this study shows that its population growth is increased by the seed dispersal by frugivorous animals compared to seed dispersal by gravity, especially when trees need to colonize forest gaps." (line 112-114)

L110: what are you referring to "This indicates that seed dispersing animals are redundant..."? Previous sentence refers to interaction frequency. Please state the indicators you used to determine that "animal species are redundant as seed dispersers".

We referred to the way animal species dispersed the seeds, i.e. the quality of seed dispersal. However, we now make this clearer by stating:

"Our results support the notion that seed-dispersing animal species in temperate forests may be largely redundant in the quality of the services they provide. Seed dispersers could potentially take over the ecological role of other species if they quantitatively compensate lost interactions." (line 116-119)

Results and Discussion

L203: Please clarify "animal seed dispersal in plants of this species..."

We now state more cautiously:

*“Our results show that animal seed dispersal can be especially beneficial for populations of *F. alnus* if new habitats are colonized. However, our findings may even underestimate the value of animal seed dispersal for the long-term persistence of plants in dynamic forests^{63,64}. We studied the population dynamics of *F. alnus* only over the course of three years and the canopy structure was held constant in the IPM. When forest succession leads to the competitive exclusion of *F. alnus* by late-successional trees, seed dispersal by animals to newly created forest gaps is essential for the long-term persistence of *F. alnus*. The results confirm the conclusions based on natural history observations about the importance of the seed dispersal mutualism between plants and animals for population persistence and forest dynamics⁶.”* (line 213-221)

L248: in what other type of habitats? please specify this.

To clarify what habitats we refer to, we now restructured the text:

“In contrast, in environments that result in very heterogeneous seed or seedling survival, the effectiveness of seed dispersal can depend strongly on the few animal species that disperse seeds to favorable microhabitats²⁷. For example, in arid environments such as deserts, most seedlings survive only in the shade of nurse plants where they are protected from heat stress⁶⁵, while seeds dispersed by gravity or deposited by animals in open areas do not contribute to plant regeneration²⁷ (but see^{66,67}).” (line 261-266)

In addition, in the following paragraph, we also state:

“It has been frequently suggested that seed dispersal by animals is especially important for escaping the increased mortality close to conspecific plants^{22,69,70}. As animals differ in their behavior and physiology, some animal species deposit seeds more often beneath conspecific plants than others, resulting in pronounced differences in seed dispersal quality among animals^{71,72}.” (line 267-271)

L298: had you find any difference about birds and mammals in their ability to compensate seed dispersal functions?

Interesting point. We were able to separate the seed dispersal function of *Martes martes* from that of other bird and mammal dispersers. We show in Fig. 1c that the seed deposition by *M. martes* does not strongly differ from that of the other dispersers. However, Perea et al. (2013) have shown that mammals usually predate/break seeds of plants during the consumption of fruits. As a consequence, we would expect that their realised qualitative function for local population growth was somewhat lower than that of most bird species (but see the seed predator *C. coccothraustes*, Fig. 3b). To clarify this, we now state:

“This was due to the small differences in the quality of seed dispersal among mammalian and avian dispersers observed before.” (line 315-316)

L331: you used the term "common" throughout the manuscript, also in its title. However, I believe it will be important to clarify what you considered as a "common seed disperser", is

this related with its abundance or frequency of interactions? Please include this aspect in the text.

We now make clear that by ‘common dispersers’ we mean those that most frequently interact with *F. alnus* which, however are also the most abundant species:

“Thus, the common bird species (e.g., S. atricapilla and T. merula), which are among the most abundant bird species in European temperate forests ⁷⁵ and most frequently interact with F. alnus, have the strongest impact on its population growth” (line 296-298)

Methodology

L378: have you applied the same methodology for surveying birds and mammals? indicate this in text.

Yes, we did. To make this clearer, we now refer to ‘frugivorous mammals and birds’ instead of only ‘frugivores’ in the text, see:

“Frugivorous mammals and birds visiting these individuals were observed with binoculars from camouflaged tents on three separate days for a period of 6 h starting at sunrise, [...]” (line 404-406)

Reviewers' comments:

Reviewer #2 (Remarks to the Author):

I have revised this manuscript for a second time and I feel the authors have successfully addressed all concerns. I recommend this study for publication.

I have posed some additional minor comments that may improve the manuscript quality.

In line 129 there is some error in a reference "⁵⁴⁵⁴".

Line 168: I can't seem to find the model results where authors test the effect of canopy cover on seedling recruitment in any of the supplementary figures or tables mentioned. However, later, in lines 258-261 appears a (contrasting?) mention to different effects of canopy cover on seedling recruitment.

Line 172: in the Suppl. Fig 6 a-d, the effect of plant size seems to differ very little among years of study, where confidence intervals have big overlaps. Also the magnitude and direction of the effects (estimates) for the models in Suppl. tables 4-5 are not reported and only Wald-X2 and p-values are reported, I find the former very informative, although its true the effects are later visible in Suppl. Fig 6.

References to the supplementary material in some parts of the results are very broad, referencing several tables and figures together (e.g. lines 170, 172, 238). It would be helpful if references to supplementary material were more specific to the highlighted results, so the reader can go the precise figure(s) or table(s).

I couldn't find the value for the inhibition factor (Frecfruit) used for seedling recruitment from seeds within fruits.

I realised during the first revision that the effectiveness landscape depicted in fig.3b is inverted as how it is normally represented, where the quantity component is normally present in x-axis and quality in y-axis, also it does not present isolines with final SDE values. I supposed this was authors' stylistic preference, since as it is now, the graph is correct and perfectly comprehensible. Mentioning it here just in case authors wanted to use the classical representation.

In Suppl. table 1, two columns say "keimung" - change to the english word.

In Suppl. table 3 - is it not clear to me why are there 17 visits for the non-observed species, please explain in the caption. Is it the mean number of visits of rare animals observed?

Finally, authors may find interesting a manuscript we recently published that backs up the main finding of this study: common dispersers contributing most to dispersal through the quantitative component (fruit consumption), varying little in their quality as dispersers. See: <https://doi.org/10.1111/ele.14141>. I think you might find it relevant for your study, but please don't feel obliged to cite it.

I congratulate the authors for the great work on the manuscript.^[1]_[SEP]

Reviewer #3 (Remarks to the Author):

My specific comments on the revision of the manuscript entitled "Common seed dispersers contribute 1 most to the persistence of a fleshy-fruited tree" are listed below.

Introduction

L49: I would recommend expressing "the plants regeneration cycle"

L97: please indicate whether observations were made by using cameras or by direct observations.

Moreover, in line 100, more specific explanation about methodology of scats sampling is needed. For example, how many scats per species were collected? I understand that you included some information in the supplementary information, however, you can include some key words in the manuscript that helps to understand the reading and the methodology applied in the study.

L104: where have you sown the seeds? In the field, in experimental pots?

In line 106 you expressed that you performed the studies over 3 years, and then stated that you have data for over 10 years. This information is confusing. Please, express this more clearly.

Results and Discussion

L129: please check the symbols in this line

L129-130: why did you assume this?

L252: please, explain how you simulated different seed deposition patterns by animal species.

Please note: In the point-by-point response to reviewers, our **answers to comments are given in red**, *quotations from the manuscript in italic*, line numbers refer to the revised manuscript without changes tracked).

Reviewers' comments:

Reviewer #2 (Remarks to the Author):

I have revised this manuscript for a second time and I feel the authors have successfully addressed all concerns. I recommend this study for publication.
I have posed some additional minor comments that may improve the manuscript quality.

In line 129 there is some error in a reference “5454”.

We corrected the mistake.

Line 168: I can't seem to find the model results where authors test the effect of canopy cover on seedling recruitment in any of the supplementary figures or tables mentioned. However, later, in lines 258-261 appears a (contrasting?) mention to different effects of canopy cover on seedling recruitment.

In Line 168, we refer to ‘seedling recruitment’ as the percentage of seedlings recruiting from seeds sown in the field. Here, we correctly provided this information in the Supplementary Discussion 1 where we state:

*“Peak recruitment of *F. alnus* was low in the Białowieża Forest and was only influenced by year ($\chi^2 = 11.32$, $p = 0.004$, Fig S5a), but not canopy cover ($\chi^2 = 0.69$, $p = 0.407$) and the interaction of year and canopy cover ($\chi^2 = 0.46$, $p = 0.795$).”*

However, later, we write ‘seedling recruitment’ and refer to both, the recruitment from seeds and first-year survival. We agree that this is confusing. We now explicitly state recruitment and survival here:

“The effect of differences in canopy cover on seedling recruitment and survival was rather small in comparison to [...].” (line 258-259)

Line 172: in the Suppl. Fig 6 a-d, the effect of plant size seems to differ very little among years of study, where confidence intervals have big overlaps. Also the magnitude and direction of the effects (estimates) for the models in Suppl. tables 4-5 are not reported and only Wald-X2 and p-values are reported, I find the former very informative, although its true the effects are later visible in Suppl. Fig 6.

We did not provide any estimates in Supplementary table 5, as we instead show the predicted relationships of those models in Supplementary figure 6. We think this is sufficient. In addition, the uncertainty is relatively high (1) as we had problems of relocating plant individuals in the field and (2) due to measurement error. Such a large uncertainty is common in demography studies of trees (see e.g. Zuidema et al. 2010, <https://doi.org/10.1111/j.1365-2745.2009.01626.x>). At best, tree growth is measured across long periods (≥ 5 yrs), but this is not feasible with most funding schemes nowadays. Accordingly, we did not change anything here.

References to the supplementary material in some parts of the results are very broad, referencing several tables and figures together (e.g. lines 170, 172, 238). It would be helpful if

references to supplementary material were more specific to the highlighted results, so the reader can go to the precise figure(s) or table(s).

Indeed, this is not very nice. We now refer to Supplementary figures more explicitly throughout the text, and restructured the Supplement materials accordingly. This led to changes in the lines 167-172, 180, 236, 238, 254 and 273.

I couldn't find the value for the inhibition factor (f_{recfruit}) used for seedling recruitment from seeds within fruits.

We now state:

"[...] f_{recfruit} is the factor by which f_{recruit} is inhibited if seedlings are recruiting from seeds within a fruit (-70%, see ¹¹), i.e. when fruits are not eaten but fall or are dropped beneath parental trees, [...]." (line 638-640)

I realised during the first revision that the effectiveness landscape depicted in fig.3b is inverted as how it is normally represented, where the quantity component is normally present in x-axis and quality in y-axis, also it does not present isolines with final SDE values. I supposed this was authors' stylistic preference, since as it is now, the graph is correct and perfectly comprehensible. Mentioning it here just in case authors wanted to use the classical representation.

In Suppl. table 1, two columns say "keimung" - change to the english word.

Changed as suggested.

In Suppl. table 3 - is it not clear to me why are there 17 visits for the non-observed species, please explain in the caption. Is it the mean number of visits of rare animals observed?

Yes, exactly. We had added asterisks for special information. For example, we wrote at the bottom of Suppl. Table 3:

*"**Fruit removal and fruit handling of these animals was not observed. We assigned them the mean values of fruit removal and handling probabilities of other rare animals in the seed deposition network (less than ten scats in total; see methods)."*

Finally, authors may find interesting a manuscript we recently published that backs up the main finding of this study: common dispersers contributing most to dispersal through the quantitative component (fruit consumption), varying little in their quality as dispersers. See: <https://doi.org/10.1111/ele.14141>. I think you might find it relevant for your study, but please don't feel obliged to cite it.

We are aware of this very recent and nice study. As it fits the scope of this manuscript, we added the reference in line 361.

I congratulate the authors for the great work on the manuscript.

Thank you very much!

Reviewer #3 (Remarks to the Author):

My specific comments on the revision of the manuscript entitled "Common seed dispersers contribute most to the persistence of a fleshy-fruited tree" are listed below.

Introduction

L49: I would recommend expressing “the plants regeneration cycle”

Changed as suggested.

L97: please indicate whether observations were made by using cameras or by direct observations.

We now state:

“[...] in a first step, we observed with binoculars the removal of fruits, and the fruit handling behavior (i.e., removing or dropping fruits, and crushing seeds) by animal species over 936 h on 52 reproductive individuals of F. alnus (Fig. 1b)^{51,52}.” (line 97-99)

Moreover, in line 100, more specific explanation about methodology of scats sampling is needed. For example, how many scats per species were collected? I understand that you included some information in the supplementary information, however, you can include some key words in the manuscript that helps to understand the reading and the methodology applied in the study.

This introductory paragraph is only a short summary of the methods. Adding very specific details about the sample size of single animal species here is too early. Instead, we provide this information later in the manuscript in the section that specifically deals with the results of the seed deposition (line 137-148). In addition, we can also discuss the associated problems of the partially low sample size in the respective paragraph. Thus, we feel that adding this information in the beginning is not necessary and would even weaken the red line of the manuscript.

L104: where have you sown the seeds? In the field, in experimental pots?

Here, we now state:

“In a fourth step, we investigated seedling recruitment by experimentally sowing 2,500 seeds in the forest¹⁵.” (line 104)

In line 106 you expressed that you performed the studies over 3 years, and then stated that you have data for over 10 years. This information is confusing. Please, express this more clearly.

Here, we now state:

“Our 10 years of research on seed dispersal in one ecosystem allowed us to seek answers to the following questions: [...]” (line 107-108)

Results and Discussion

L129: please check the symbols in this line

Changed.

L129-130: why did you assume this?

We needed to harmonize the information by different sample techniques. To make this link clearer, we now state:

“To harmonize the complementary seed disperser information obtained by the different methods⁵⁴, we assumed that the effects of these eight animal species were functionally similar to those of other rare dispersers of F. alnus for which data from seed removal observations were available (see methods).” (line 128-131)

L252: please, explain how you simulated different seed deposition patterns by animal species.

We describe in the next sentence how we simulated different seed deposition patterns by animal species. To make clearer that it is based on a simulation, we now state:

“To test the sensitivity of seed dispersal quality to variation in seed deposition, we simulated unrealistic extreme patterns of seed deposition. For example, we simulated that animals dispersed seeds exclusively to bright environments (Supplementary Figure 12).” (line 251-254)

Thank you for your review.